# Deglacial export of pre-aged terrigenous carbon to the Bay of Biscay

Eduardo Queiroz Alves[1,†], Wanyee Wong[1,2,‡], Jens Hefter[1], Hendrik Grotheer[1,4], Tommaso Tesi[3], Torben Gentz[1], Karin Zonneveld[4], and Gesine Mollenhauer[1,2,4]

[1]Alfred Wegener Institute for Polar and Marine Research, Bremerhaven, Germany
[2]University of Bremen, Bremen, Germany
[3]Institute of Polar Sciences - National Research Council, Bologna, Italy
[4]MARUM-Center for Marine Environmental Sciences, University of Bremen, Bremen, Germany
[†]now at Departamento de Geoquímica, Universidade Federal Fluminense, Brazil
[‡]now at NORCE Norwegian Research Centre, Bjerknes Centre for Climate Research, Bergen, Norway

**Correspondence:** Eduardo Queiroz Alves (eduardoa@id.uff.br)

**Abstract.** The last deglaciation is the most recent relatively well-documented period of pronounced and fast climate warming and, as such, it holds important information for our understanding of the climate system. Notably, while research into terrestrial organic carbon reservoirs has been instrumental in exploring the possible sources of atmospheric carbon dioxide during periods of rapid change, the underlying mechanisms are not fully understood. Here we investigate the mobilization of organic matter to the Bay of Biscay, located in the northeastern Atlantic Ocean off the coasts of France and Spain. Specifically, we focus on the area that was the mouth of the Channel River during the last deglaciation, where an enhanced terrigenous input has been reported for the last glacial-interglacial transition. We conducted a comprehensive suite of biomarker analyses (e.g., $n$-alkanes, hopanes, and $n$-alkanoic acids) and isotopic investigations (radiocarbon dating and $\delta^{13}$C measurements) on a high-resolution sedimentary archive. The present study provides the first direct evidence for the fluvial supply of immature and ancient terrestrial organic matter to the core location. Moreover, our results reveal the possibility of permafrost carbon export to the ocean, driven by processes such as deglacial warming and glacial erosion. These findings are consistent with observations from other regions characterized by present or past permafrost conditions on land, which have shown that permafrost thaw and glacial erosion can lead to carbon remobilization, potentially influencing atmospheric carbon dioxide levels.

## 1 Introduction

High-latitude permafrost soils hold ca. 1000 Pg of soil organic carbon (C) in the upper 3 m of the northern circumpolar region (Hugelius et al., 2014). This immense amount of C is in the form of frozen organic matter (OM), the thawing of which releases greenhouse gases, inducing positive feedback mechanisms that have implications for the C cycle on a global scale (Zimov et al., 2006; Schuur et al., 2008, 2009; Hugelius et al., 2014; Schuur et al., 2015). While current climate change raises concerns about the stability of these massive pools of organic C (Vonk et al., 2012; Schneider Von Deimling et al., 2015), the dynamic character of Earth's climate means that past trends and variability can be examined to improve future projections of this effect. The hypothesis of a combined contribution of ancient terrestrial C, potentially derived from thawing permafrost, and marine C sources to elevated atmospheric levels of carbon dioxide ($CO_2$) and methane ($CH_4$) during the last deglaciation is discussed in

several studies (Ciais et al., 2012; Köhler et al., 2014; Bauska et al., 2016; Crichton et al., 2016; Simmons et al., 2016). Over the course of the Last Glacial Maximum (LGM), large expanses of continuous permafrost were found in the Eurasian continent,

covering much of central and western Europe, in areas where permafrost cover today no longer exists (Vandenberghe and Pissart, 1993; Levavasseur et al., 2011; Vandenberghe et al., 2012; Žák et al., 2012; Schaefer et al., 2014; Vandenberghe et al., 2014) (Figure 1). These regions comprise the southern edge of the LGM permafrost area and, according to Köhler et al. (2014), are likely to have experienced a rapid loss of massive amounts of ancient C as a result of thawing during the last deglaciation. However, while direct evidence for the deglacial remobilization of ancient C from permafrost has been reported for the Arctic

(Tesi et al., 2016; Keskitalo et al., 2017; Martens et al., 2019, 2020; Wu et al., 2022) and subarctic (Winterfeld et al., 2018; Meyer et al., 2019), similar data are still lacking for the European realm where the phenomenon has been suggested on the basis of enhanced terrigenous biomarker concentrations in sediment cores (Ménot et al., 2006; Rostek and Bard, 2013; Soulet et al., 2013).

During the LGM, continental glaciers were part of the European landscape. The Fennoscandian (FIS) and the British-Irish

(BIIS) ice sheets covered most of Britain, Ireland, Northern Europe and the North Sea (Bowen et al., 2002; Svendsen et al., 2004; Mangerud et al., 2004), contributing to the lower eustatic sea level and altering coastlines (e.g., Fairbanks, 1989; Lambeck, 1997; Lambeck et al., 2014). This sea-level lowstand, paired with the configuration of the BIIS and the FIS, led to a reorganization of major European drainage basins, with continental runoff being funnelled through the English Channel (Gibbard, 1988). The so-called Fleuve Manche or Channel River received the runoff of major European rivers, carrying meltwaters

from glaciers and ice sheets (e.g., Antoine et al., 2003; Bourillet et al., 2003) (Figure 1). As a consequence, changes in the hydrological cycle in Europe during the last glacial-interglacial transition induced a strong response from this system, resulting in increased water flow in the Channel River and its tributaries (e.g., Ménot et al., 2006; Toucanne et al., 2009, 2010). Permafrost developed in the glacier-free areas of the continent, extending across a large portion of the Channel River (Figure 1), and towards the end of the last glaciation it is likely to have reached its maximum extent, with discontinuous permafrost present

in regions almost as far south as the Mediterranean Sea (Vandenberghe et al., 2014).

Permafrost encompasses a diverse range of organic-rich deposits, including ancient peat, organic-rich soils, and potentially mineral soils, all of which can become stabilized (frozen) under permafrost conditions, preserving C within them. There is evidence for the presence of several peatlands with active peat deposition in northern latitudes (> 40°N), including Northern Europe, during the last interglacial (130 - 116 kyr BP; Treat et al., 2019). Although the occurrence of permafrost during the

LGM likely resulted in the long-term burial of peatland OM, deglacial permafrost thawing may have led to the fluvial export of peat-derived OM to the ocean (see e.g., Schefuß et al., 2016; Garcin et al., 2022). In addition to permafrost, petrogenic material carried by glacial meltwater may have been another source of fossil OM to the oceans during the last deglaciation. Following glacial retreat, the mechanical erosion of bedrock such as oil shales mobilizes petrogenic C, which is transported as finely ground glacial meal to the oceans (Koppes and Hallet, 2002, 2006). Therefore, if we are to accurately quantify the

impact of the permafrost C feedback on Earth's climate, it is imperative to distinguish between the possible origins of the OM deposited on the continental margins over the course of the last deglaciation. Considering the presence of shale formations in Europe (e.g., Zhao et al., 2022), our study explores two primary hypotheses regarding the origin of OM: permafrost-derived

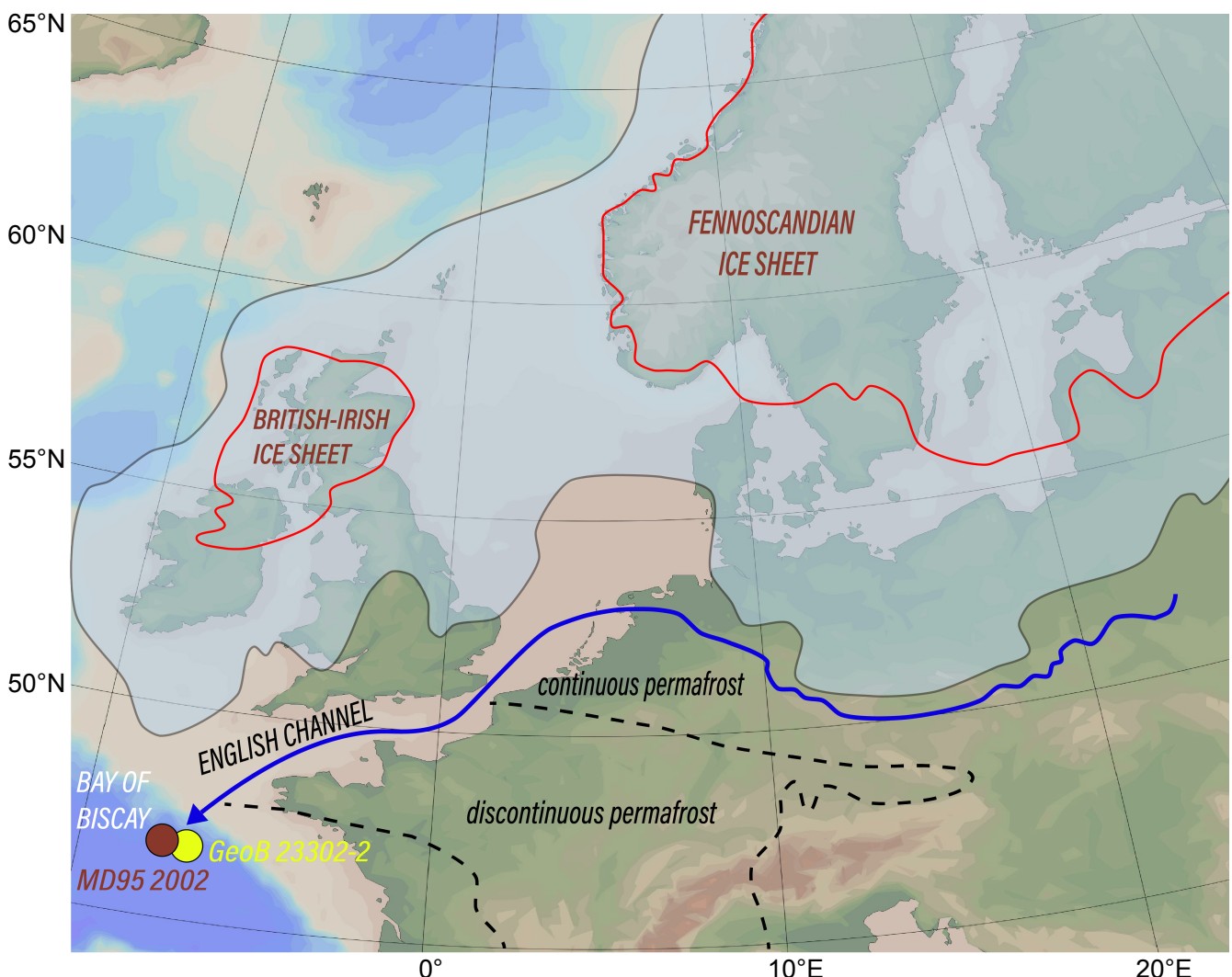

**Figure 1.** Northwest Europe during the LGM. The blue arrow indicates the downstream course of the Channel River from the eastern flank of the FIS. The dashed lines show the distribution of permafrost (Renssen and Vandenberghe, 2003, and references therein) and the red contours indicate the approximate limits of the ice sheets at ca. 17 ka BP (Patton et al., 2017). The yellow dot illustrates the location where core GeoB23302-2, used in the present study, was retrieved. Map based on that in Ménot et al. (2006), who studied core MD95 2002 from a nearby location (red dot).

OM, which includes peat preserved in past permafrost-covered European regions, and OM originating from petrogenic sources. Here, we analyzed organic biomarkers and conducted compound-specific radiocarbon ($^{14}$C) measurements on *n*-alkanoic acids isolated from a high-resolution, well-dated marine sediment core retrieved from the Channel River outflow to evaluate these hypotheses.

## 2 Materials and methods

Herein, we provide an overview of our analytical approach; for in-depth descriptions of each method and the specific details of our analyses, please refer to the following subsections. All the elemental, isotopic and biomarker analyses described in this section were conducted as part of the present study. In this research we used several analytical tools to examine core GeoB23302-2 (47°26.61'N, 8°28.67'W; 2167 m water depth) (Figure 1). The chronology of the sedimentary sequence was established using an age-depth model constructed with the $^{14}$C ages of planktic foraminifera in the OxCal software (Bronk Ramsey, 1995, 2009a). Elemental ratios were obtained through X-ray fluorescence (XRF) core scanning. The coarse fraction of sediments tends to be enriched in zirconium (Zr), while rubidium (Rb) is found in fine-grained minerals. Therefore, here we report the ratio Zr/Rb as an elemental measure of grain size, which has been used as a proxy for river runoff (Dypvik and Harris, 2001; Kylander et al., 2011; Wang et al., 2011; Wu et al., 2020). Similarly, given that iron (Fe) is normally associated with continental weathering products and that the calcium (Ca) content in the sediment primarily reflects the presence of marine carbonate, here we use the Fe/Ca ratio as a provenance indicator, reflecting variations in terrigenous sediment delivery (Arz et al., 1999; Itambi et al., 2009; Dickson et al., 2010; Perez et al., 2016). We analysed lipid biomarkers using solvent extraction and gas chromatography and calculated $n$-alkane-derived indices, namely the carbon-number preference index (CPI$_{alk}$) (e.g., Bray and Evans, 1961; Marzi et al., 1993) and the proxy ratio P$_{aq}$ (Ficken et al., 2000), which are commonly used in environmental investigations (e.g., Nichols et al., 2006; Rommerskirchen et al., 2006; Zhou et al., 2012; He et al., 2020; Feurdean et al., 2021) to assess the degree of OM degradation and reconstruct the temporal evolution of continental vegetation systems, respectively. The CPI$_{alk}$ is based on the ratio of odd to even $n$-alkanes, providing information about the distribution of these compounds in the samples, while the P$_{aq}$ reflects the predominance of long-chain $n$-alkanes in terrestrial vascular plants as opposed to algae and macrophytes, which primarily synthesize short- to mid-chain $n$-alkanes (Bianchi and Canuel, 2011). The P$_{aq}$ ratio indicates the relative input of aquatic macrophytes and terrestrial plants to the sediment (Ficken et al., 2000). The prevalence of odd-numbered $n$-alkanes in fresh material implies that the CPI$_{alk}$ can serve as an indicator of OM degradation (Bray and Evans, 1961; Marzi et al., 1993; Meyers and Ishiwatari, 1993). To further assess the presence of petrogenic OM, we used the fractional abundance of hopanes of biological origin, e.g., bacteria-derived hopanes, in relation to their diagenetic isomers (f$\beta\beta$) (Meyer et al., 2019). Here, the branched and isoprenoid tetraether (BIT) index (Hopmans et al., 2004), based on the relative abundance of branched glycerol dialkyl glycerol tetraether lipids (GDGTs) characteristic of terrestrial bacteria and the isoprenoid GDGT crenarcheol produced by marine Thaumarchaeota, is used as a proxy for the input of terrestrially-sourced OM. Carbon isotope analysis ($\delta^{13}$C) of bulk samples was conducted as part of our investigation into their origin. Finally, given that the results of bulk $^{14}$C dating reflect the $^{14}$C content of a heterogeneous mixture of compounds possibly derived from distinct sources, here we further address the provenance of the OM by using compound-specific $^{14}$C analyses of high molecular weight $n$-alkanoic acids (C$_{26:0}$, C$_{28:0}$ and C$_{30:0}$) - which are typically derived from vascular plants (Bianchi and Canuel, 2011) - from specific depths in the core. This approach has been successfuly employed for the identification of ancient terrigenous material export at other sites (e.g., Winterfeld et al., 2018; Meyer et al., 2019; Wu et al., 2022).

## 2.1 Sampling and core chronology

Core GeoB23302-2 was recovered from the Celtic Margin, off the English Channel (47°26.61'N, 8°28.67'W; 2167 m water depth) (Figure 1), with the help of a gravity corer during cruise MSM 79 of the research vessel Maria S. Merian. The core location is in close proximity to the site where core MD95 2002, which has been studied in previous publications (e.g., Ménot et al., 2006; Toucanne et al., 2015), was retrieved (47°27'N, 8°32'W) (see Figure 1). The chronology of our 700 cm core was established based on seven radiocarbon accelerator mass spectrometry ($^{14}$C-AMS) measurements of planktic foraminifera (*G. bulloides* and *N. pachyderma*) picked at specific depths (Table 1). The preparation and measurement of these samples followed well-established protocols routinely run at the MICADAS $^{14}$C laboratory of the Alfred Wegener Institute (AWI) (Mollenhauer et al., 2021). The $^{14}$C ages were uploaded to the OxCal software version 4.4.2 (Bronk Ramsey, 1995, 2009a), and assuming that the deposition is a Poisson process, the P Sequence model was employed for the construction of an age-depth model for core GeoB23302-2 (Bronk Ramsey, 2008; Bronk Ramsey and Lee, 2013). This deposition model (Figure 2 in the Supplementary Material) uses the global marine calibration curve Marine20 (Heaton et al., 2020), and a local marine reservoir correction $\Delta$R of $94 \pm 45$ $^{14}$C yr (Tisnérat-Laborde et al., 2010). It is important to note that while Marine20 incorporates larger marine reservoir age estimates for the last glacial period than Marine13, these estimates are considered more realistic due to methodological improvements in the former (Heaton et al., 2020, 2023). A general outlier analysis was employed to account for possible outliers within the chronological model (Bronk Ramsey, 2009b). The code is available in the supplementary material accompanying this paper.

## 2.2 Elemental analyses

The XRF characterization of core GeoB23302-2 was performed using the XRF Core Scanner II (AVAATECH Serial No. 2) at the Center for Marine Environmental Sciences (MARUM), University of Bremen, Germany. Measurements were performed at 1 cm intervals for the upper 3.5 m of the core and at every 2 cm for the remaining section. The scan resolution was set to 1 cm with 2 running rounds, during which the elements were detected with 10 and 30 kV of tube voltage. In order to account for the closed sum effects of water content, grain size and OM amount (e.g., Weltje and Tjallingii, 2008), we report the elemental ratios Zr/Rb and Fe/Ca.

## 2.3 Biomarker analyses and derived indices

Sediment samples taken at 10 cm intervals from core GeoB23302-2 were freeze-dried and homogenized. For each depth, approximately 3 g of sediment were subsampled and underwent ultrasonic extraction with a mixture of dichloromethane:methanol 9:1 (v:v). This step was repeated three times and the total lipid extracts obtained were then saponified with 0.1 M potassium hydroxide (KOH) in methanol:water 9:1 at 80 °C for 2 h. This procedure resulted in the separation of the neutral lipids and *n*-alkanoic acids fractions, which were subsequently extracted using *n*-hexane and dichloromethane (at pH 1), respectively. Next, silica gel chromatography was employed to further split the neutral lipids via elution with *n*-hexane and dichloromethane:methanol 1:1 (v:v), yielding the *n*-alkanes and GDGTs subfractions, respectively. The *n*-alkane concentra-

tions were measured via gas chromatography (GC) using a 7890A GC (Agilent Technologies) equipped with a flame ionization detector (FID) and DB-5MS fused silica capillary columns (60 m, ID 250 $\mu$m, 0.25 $\mu$m film coupled to a 5 m, ID 530 $\mu$m deactivated fused silica precolumn). Retention times and the comparison with an *n*-alkane standard were used for the identification

of different compounds whereas quantifications were achieved through the use of an internal standard (squalane) added to the sample prior to extraction. We calculated *n*-alkane-derived indices, namely the CPI$_{alk}$ (e.g., Bray and Evans, 1961; Marzi et al., 1993):

$$CPI_{alk} = \frac{1}{2} \cdot \left( \frac{C_{25} + C_{27} + C_{29} + C_{31} + C_{33}}{C_{24} + C_{26} + C_{28} + C_{30} + C_{32}} + \frac{C_{25} + C_{27} + C_{29} + C_{31} + C_{33}}{C_{26} + C_{28} + C_{30} + C_{32} + C_{34}} \right), \tag{1}$$


and the P$_{aq}$ (Ficken et al., 2000):

$$P_{aq} = \frac{C_{23} + C_{25}}{C_{23} + C_{25} + C_{29} + C_{31}}, \tag{2}$$


Hopanes were analyzed via GC coupled with time of flight mass spectrometry (GC-TOF-MS) and such a system consisted of a LECO Pegasus III (LECO Corp., St. Joseph, MI) interfaced to an Agilent 6890 GC which was equipped with a temperature programmable cooled injection system (CIS4, Gerstel). The measurements were performed using the instrumental setup described in Hefter (2008) and identification was achieved through the relative retention times and mass spectra. The sum of

m/z 191 and 205 was used for the quantification of homohopane isomers (C$_{31}$), namely the 17$\beta$,21$\beta$ (H), 22R homohopane, the 17$\beta$,21$\alpha$ (H), 22R + 17$\beta$,21$\alpha$ (H), 22S homohopanes, the 17$\alpha$,21$\beta$ (H), 22R homohopane, and the 17$\alpha$,21$\beta$ (H), 22S homohopane. Next, the f$\beta\beta$ was calculated (Meyer et al., 2019):

$$f\beta\beta = \frac{C_{31}\beta\beta R}{C_{31}\beta\beta R + C_{31}\alpha\beta S + C_{31}\alpha\beta R + C_{31}\beta\alpha S + C_{31}\beta\alpha R} \tag{3}$$


The analyses of branched and isoprenoid GDGTs by High Performance Liquid Chromatography (HPLC) were performed on an Agilent 1200 series HPLC system coupled to an Agilent 6120 single quadrupole MS via an atmospheric pressure chemical ionization interface (APCI), broadly following the method described in Hopmans et al. (2016). The chromatographic separa-

tion of individual GDGTs was achieved via the use of two UPLC silica columns in series (Waters Acquity BEH HILIC, 2.1

mm x 150 mm, 1.7 $\mu$m and a 2.1 mm x 5 mm pre-column of the same material) maintained at 30 °C. Positive-ion APCI-MS and selective ion monitoring (SIM) of (M+H)$^+$ ions (Sinninghe Damsté et al., 2000) or ion-source fragmentation products of OH-GDGTs (Liu et al., 2012) allowed the identification of GDGTs. Quantification was performed with the use of an internal standard (C$_{46}$-GDGT) added prior to extraction. For this research, we calculated the BIT index (Hopmans et al., 2004):


$$BIT = \frac{I + II + III}{I + II + III + cren} \tag{4}$$

where the roman numerals refer to specific GDGTs characteristic of terrestrial bacteria and cren stands for crenarchaeol, which
is derived from marine planktonic Thaumarchaeota.

## 2.4    Compound-specific radiocarbon analyses (CSRA)

Soxhlet extraction was employed for the compound-specific $^{14}$C dating of high molecular weight $n$-alkanoic acids. For that purpose, approximately 100 g of freeze-dried and homogenized sediment taken from selected depths in core GeoB23302-2 were extracted for 48 h using a mixture of dichloromethane:methanol 9:1 (v:v). Total lipid extracts were saponified with
0.1 M KOH in methanol:water 9:1 at 80 °C for 2 h and the $n$-alkanoic acids were recovered from the saponified solution using $n$-hexane at pH 1. Next, $n$-alkanoic acids were methylated at 80 °C overnight in a nitrogen atmosphere with HCl and methanol of known $^{14}$C signature to yield the fatty acid methyl esters (FAMEs) that were later extracted with $n$-hexane. Silica gel chromatography was employed to separate FAMEs from polar compounds. The $n$-C$_{26:0}$, $n$-C$_{28:0}$ and $n$-C$_{30:0}$ alkanoic acids underwent purification via preparative capillary GC (PC-GC; Eglinton et al., 1996) on an Agilent HP6890N GC with
a Gerstel Cooled Injection System (CIS) connected to a Gerstel preparative fraction collector (Kusch et al., 2010). A Restek Rxi-1ms fused silica capillary column (30 m, 0.53 mm diameter, 1.5 $\mu$m film thickness) equipped the GC. Injection was performed stepwise with 5 $\mu$L per injection and, at the end of the process, the purity of the FAMEs was checked by analyzing aliquots of the samples via GC-FID. The purified FAMEs were transferred to tin capsules (25 $\mu$L volume; ELEMENTAR) using dichloromethane, dried on a hot plate at 40 °C and packed. An Elementar vario ISOTOPE EA (Elemental Analyzer)
was used for the combustion of the samples, generating CO$_2$ with carbon isotopic ratios directly determined by the connected MICADAS system. Reference standards (oxalic acid II; SRM 4990C) and $^{14}$C-free CO$_2$ gas had their $^{14}$C content measured together with the samples. The BATS software (Wacker et al., 2010) was used for blank corrections and standard normalization and the final results are reported as fraction modern carbon (F$_m$).

## 2.5    Assessment and correction of CSRA procedure blank

The preparation procedures for CSRA introduce exogenous C, i.e., contaminants, to samples. The degree of contamination varies according to the methods employed and, in our case, processes such as column bleed and carry-over during prep-GC

compound isolation may contribute to this. For this reason, assessing the $F_m$ and the size of the blank ($F_{mblank}$ and $m_{blank}$, respectively) is essential for accurate results. Here, in-house reference samples of $^{14}$C-free Messel Shale ($F_m = 0$) and modern apple peel ($F_m = 1.029 \pm 0.001$) underwent the same pre-treatment as samples of unknown age and their results were used for

blank correction following the method outlined in Sun et al. (2020). Isotopic mass balance was employed in order to make a correction for the methyl group added during the derivatization of the samples. Uncertainties were fully propagated.

## 2.6   Pre-depositional $^{14}$C ages of terrigenous compounds

The $\Delta^{14}$C values of the *n*-alkanoic acids analysed here were corrected for radioactive decay between 1950 and 2021, which is the year of measurement. These values were then used to calculate the $\Delta^{14}$C values at the time of deposition:

$$\Delta^{14}C_{initial} = \left[ \left( \frac{\Delta^{14}C}{1000} + 1 \right) \cdot e^{\lambda t} - 1 \right] \cdot 1,000 \tag{5}$$

where $\lambda$ is a decay constant (1/8,267 yr$^{-1}$) and t is the time of deposition. The $\Delta^{14}$C values of the atmosphere contemporaneous with the compounds ($\Delta^{14}C_{atm}$) were obtained from comparison with the IntCal20 dataset (Reimer et al., 2020) using the age ranges given by the deposition model for the respective sediment layers. Finally, pre-depositional $^{14}$C ages for the *n*-alkanoic acids were given by:

$$A = -8,033 \cdot ln \left( \frac{1 + \Delta^{14}C_{initial}/1,000}{1 + \Delta^{14}C_{atm}/1,000} \right) \tag{6}$$

These calculations follow the method outlined in Schefuß et al. (2016) and later in Winterfeld et al. (2018), where more details can be found.

## 2.7   Stable isotope analyses

Carbon stable isotope ($\delta^{13}$C) analyses were carried out on acidified samples (Ag capsules, HCl, 1.5 M) in order to remove

the inorganic C (Nieuwenhuize et al., 1994). Analyses were performed using a Thermo Scientific DELTA Q Isotope Ratio Mass Spectrometer coupled to a Thermo Scientific FLASH 2000 CHNS/O Analyzer via Conflo III at the Stable Isotope Laboratory of ISP-CNR. $\delta^{13}$C data are expressed in the conventional delta notation (‰). Isotopic data were calibrated using the IAEA reference material IAEA-CH7 polyethylene, -32.15‰vs VPDB). Throughout the runs, we used other standards with a sediment matrix routinely used in the laboratory to check the reproducibility of measurements. The standard deviation for

$\delta^{13}$C measurements was lower than $\pm 0.1$‰based on replicates of sediment standards.

## 3   Results

The GeoB23302-2 sediment core spans a period from approximately 25 to 4 cal kyr BP (Table 1). The age-depth model for this core shows a period of enhanced deposition between approximately 20 and 15 cal kyr BP (Figure 2 in the Supplementary

Material). An outlier analysis shows that the model represents well the foraminifera $^{14}$C ages, with an OxCal overall agreement index of 99%. From approximately 20.6 until 15 cal kyr BP, the Zr/Rb ratio shows a long-term increase, whereas a period of relatively high Fe/Ca values is observed at ca. 21 - 16.4 cal kyr BP (Figure 2c). Values for the proxy $P_{aq}$ are relatively low during the LGM and show a pronounced increase at approximately 21 cal kyr BP with higher values towards the late glacial and a sudden decrease at the onset of Heinrich event 1 (iceberg discharge from the Laurentide ice sheet into the North Atlantic Ocean; H1; ca. 17.2 cal kyr BP) (Figure 2d). This is followed by a sharp increase at approximately 16.5 cal kyr BP and a drop to Holocene values around 16.3 cal kyr BP. The $CPI_{alk}$ and the $f\beta\beta$ proxy show relatively low values in the late glacial/early deglaciation when compared to the Holocene (Figure 2e). During the LGM, the $CPI_{alk}$ values are broadly constant while an increase is observed at the beginning of H1 followed by a sharp drop around 16.5 cal kyr BP and a return to higher values at approximately 16.2 cal kyr BP. The $f\beta\beta$ record shows fluctuations during the LGM, followed by a gradual decrease starting at approximately 18 cal kyr BP and a sharp drop around 16.5 cal kyr BP before returning to higher values at approximately 16.2 cal kyr BP (Figure 2e). The $^{14}$C ages of the long chain *n*-alkanoic acids varied from approximately 10 to 39 $^{14}$C kyr. When converted to pre-depositional age estimates, it is possible to observe that at the peak of our BIT record, around 18 cal kyr BP, compounds pre-aged by up to ca. 25,000 $^{14}$C yr were delivered to the continental slope (Figure 2f). Pre-depositional ages broadly follow the BIT record, with younger compounds observed from the end of the BIT peak (ca. 16 cal kyr BP) towards the Holocene. The results of our bulk $\delta^{13}$C analyses corroborate the BIT index record, showing that terrestrial C dominantly contributed to samples during the period of enhanced terrigenous deposition, while the OM in Holocene samples is mostly marine (Table 2).

**Table 1.** Radiocarbon and modelled ages of planktic foraminifera picked from core GeoB23302-2.

| Depth (cm) | Species | $^{14}$C age ($^{14}$C yr) | Modelled age (cal BP; $2\sigma$) |
|---|---|---|---|
| 3-4 | *G. bulloides* | $3556 \pm 67$ | 3600 - 3100 |
| 118-119 | *G. bulloides* | $9981 \pm 105$ | 11300 - 10600 |
| 234-235 | *N. pachyderma* | $14111 \pm 124$ | 16800 - 15900 |
| 340-341 | *N. pachyderma* | $15131 \pm 142$ | 17800 - 17100 |
| 450-451 | *N. pachyderma* | $15112 \pm 151$ | 18100 - 17400 |
| 584-585 | *G. bulloides* | $16893 \pm 203$ | 20200 - 19100 |
| 686-687 | *G. bulloides* | $20432 \pm 202$ | 24200 - 23100 |

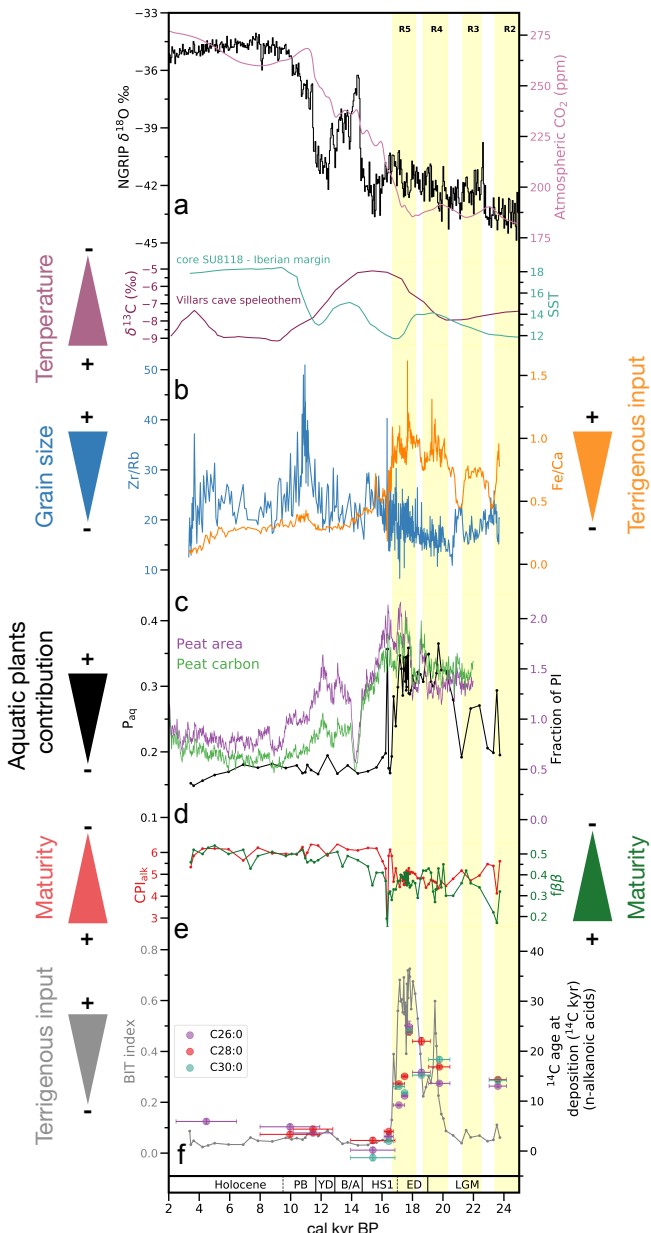

**Figure 2. a:** NGRIP $\delta^{18}$O (Andersen et al., 2004) and atmospheric CO$_2$ (Köhler et al., 2017) records. **b:** Sea surface temperatures in the Northeast Atlantic Ocean (Bard et al., 2000) and $\delta^{13}$C values from speleothems in Western Europe (Wainer et al., 2011). The data were smoothed to eliminate very high-frequency components. **c:** Zr/Rb and Fe/Ca elemental ratios from XRF data. **d:** Peatland area (purple) and carbon (green) in Europe as a fraction of pre-industrial (PI) values (0.231 Mkm$^2$ and 19.6 GtC) (Müller and Joos, 2020) and the P$_{aq}$ index record. **e:** f$\beta\beta$ and CPI$_{alk}$ records. **f:** Pre-depositional $^{14}$C ages of *n*-alkanoic acids from core GeoB23302-2 and BIT index record. Yellow bands mark major flooding events of the Channel River (Toucanne et al., 2015).

**Table 2.** Carbon stable isotope ($\delta^{13}$C) values of bulk sediment samples from core GeoB23302-2.

| Depth (cm) | Age ($^{14}$C yr) | Uncertainty ($^{14}$C yr) | $\delta^{13}$C (‰) |
|---|---|---|---|
| 30-33 | 4467 | 1988 | -22.25 |
| 90-93 | 9957 | 1960 | -22.22 |
| 140-143 | 11471 | 1304 | -22.07 |
| 202-205 | 15397 | 1456 | -23.37 |
| 247-250 | 16441 | 301 | -23.88 |
| 302-305 | 17108 | 373 | -25.72 |
| 350-353 | 17482 | 194 | -25.93 |
| 451-454 | 17778 | 236 | -25.83 |
| 510-513 | 18585 | 591 | -25.09 |
| 590-593 | 19761 | 699 | -24.44 |
| 680-683 | 23604 | 576 | -23.23 |

## 4 Discussion

### 4.1 Source of the OM in the sedimentary record

Our results corroborate the previous findings of Ménot et al. (2006), showing a notable increase in the influx of terrigenous
OM in the Bay of Biscay during the last deglacial period. Between ca. 20.6 and 15 cal kyr BP, the values of the Zr/Rb ratio
reflect the deposition of coarse-grained sediments at the core location, which may be associated with enhanced fluvial activity
(Wang et al., 2011). A period of relatively high Fe/Ca values (ca. 21 - 16.4 cal kyr BP) is indicative of a greater influx of
sediment from land (Figure 2c). This is consistent with a period of elevated terrestrial contribution to the bulk OM present
in the sediment, from approximately 20.5 to 16.5 cal kyr BP, except for a sudden decrease at approximately 19 cal kyr BP,
as revealed by the BIT index (see e.g., Hopmans et al., 2004; Herfort et al., 2006; Kim et al., 2006; Schouten et al., 2013;
Grotheer et al., 2020) (Figure 2f). These Fe/Ca and BIT patterns are also recorded in core MD95 2002 (see Figures 3 and
4 in the Supplementary Material; Toucanne et al., 2015; Ménot et al., 2006, respectively) and a similar pattern of marked
Fe/Ca peaks, sometimes associated with peaks in OM content, during Heinrich events and much lower values throughout the
Holocene has been observed at other sites (Jennerjahn et al., 2004; Lebreiro et al., 2009; Zhang et al., 2015; Crivellari et al.,
2018).

Beyond identifying the presence of terrestrial OM transported to the Bay of Biscay via the Channel River during the LGM-
Holocene transition, the comprehensive analysis of elemental, geochemical, and isotopic proxies presented here provides in-
sights into the potential sources of this terrigenous OM. Values for the P$_{aq}$ proxy point to a major contribution of OM from
aquatic plants between approximately 20.2 and 17.2 cal kyr BP, suggesting the presence of OM sourced from wetland veg-
etation (Figure 2d). Our CPI$_{alk}$ record reflects the degree of degradation the sedimentary OM has undergone in its previous
terrestrial reservoir or during transportation (cf. Bröder et al., 2018). During the peak of terrigenous deposition, the signal of

more mature OM fluvially transported to the continental slope is detected in our $CPI_{alk}$ and $f\beta\beta$ records, which reach relatively low values when compared to the Holocene (Figure 2e). The presence of petrogenic, i.e., thermally-mature, material (Farrington and Tripp, 1977; Jeng, 2006) is another factor to consider when interpreting these records. Nonetheless, throughout the archive, the $CPI_{alk}$ values remain above the diagnostic value of petrogenic material (ca. 1) reported by Bray and Evans (1961). Additionally, in contrast with the results of Meyer et al. (2019) for the Bering Sea, in the present study the $f\beta\beta$ proxy is not indicative of petrogenic material in the sediment (see the Supplementary Material). This is because the values do not decrease in response to diagenetic transformations but rather due to enhanced inputs of the $C31\alpha\beta R$ hopane, which is abundant in peat (Inglis et al., 2018).

The compound-specific $^{14}C$ results disclose that the analyzed terrigenous biomarkers are ancient and pre-aged, indicating that they are not contemporaneous with sediment deposition but rather older (Figure 2f). Despite the large variability of OM residence time in permafrost soils, the pre-depositional ages of some of the compounds present in core GeoB23302-2 (up to 25,000 $^{14}C$) are considerably greater than those previously attributed to permafrost-derived OM at other sites and at different timescales (e.g., up to ca. 10,000 $^{14}C$; Gustafsson et al., 2011; Winterfeld et al., 2018). Although petrogenic contributions are commonly thought to be devoid of *n*-alkanoic acids, this assumption does not always hold (Kvenvolden, 1966) and the erosion of organic-rich sedimentary rocks can supply fossil OM to the ocean, thereby depleting the $^{14}C$ compound-specific signal in the sediment (e.g., Raymond et al., 2004; Copard et al., 2007; Wu et al., 2022, and references therein). However, the OM present in core GeoB23302-2 presents $\delta^{13}C$ values in the range from -25.9 to -22.3‰, heavier than the average value of $\delta^{13}C$ = -29.4 ‰ displayed by organic-rich rocks in the region (e.g., Zhao et al., 2022). These values are rather comparable to those observed in peat, corroborating the results of the $CPI_{alk}$ and $f\beta\beta$ proxies and pointing to a pre-aged immature source such as ancient peat. The pre-depositional ages observed during the peak of OM deposition likely result from a $^{14}C$ reservoir effect, which is caused by radioactive decay and a lack of exchange with the atmosphere (Stuiver and Polach, 1977). This effect occurs as the OM is stored in its source reservoir before eventual erosion and transportation to the core location. Mechanisms such as deposition-resuspension loops on the continental slope could be invoked to explain pre-aged *n*-alkanoic acids during the most recent part of our record (Kusch et al., 2021, and references therein). To summarize, our results strongly support previous findings describing a massive remobilization of terrestrial C from the European continent to the North Atlantic Ocean during the last deglaciation, with notable peaks at ca. 19.5 and between approximately 19 and 17 cal kyr BP (Ménot et al., 2006). In addition, the set of proxies applied in the present study allows us to go further and suggest peat-derived material as a major source of OM to the Bay of Biscay during the last deglaciation.

## 4.2 Landscape development and OM remobilization mechanisms

Wetlands are dynamic ecosystems that fix $CO_2$ from the atmosphere, store C and contribute to the C cycle through various processes, including the decomposition of OM that releases $CO_2$ (Mitra et al., 2003). Therefore, the establishment of wetlands in the study region towards the end of the LGM and during the last deglaciation (Figure 2d), combined with permafrost distribution data that imply gradual permafrost decomposition (e.g., Vandenberghe and Pissart, 1993; Levavasseur et al., 2011; Vandenberghe et al., 2012; Schaefer et al., 2014; Vandenberghe et al., 2014) and records of atmospheric $CO_2$ concentration

(Köhler et al., 2017), suggests the need to investigate thawing permafrost as a possible source of OM to the deposition site. Although some of the peatlands formed during the Eemian period were buried by glaciers and mineral sediments, Weichselian ice sheets were not as extensive as the ones from the Saalian and some deposits remained uncovered (e.g., in Northern Germany, on the cliff of the Elbe River; Ehlers et al., 2011). Peat deposits formed during the last interglacial occur widely across the studied region (Turner, 2000), from Belgium (e.g., De Moor, 1983) and the Netherlands (e.g., Schokker et al., 2004) to Poland (e.g., Woronko et al., 2018), through Germany (e.g., Börner et al., 2018; Grube and Usinger, 2017) and Denmark (e.g., Christiansen, 1998). Although the environmental conditions of the last glaciation were unfavorable for the development of peatlands, factors such as the formation of permafrost in Europe resulted in the long-term preservation of OM from older periods (e.g., frozen peat OM) (Treat et al., 2019). However, during the last deglaciation, the thawing of permafrost and the presence of meltwater streams may have contributed to the erosion of these peats. We propose that Eemian peatlands represent the primary source of fossil biomarkers transported to the Bay of Biscay, with processes such as thermal and physical erosion of these deposits (see e.g., Sidorchuk et al., 2009, 2011) leading to pre-aged material reaching the final burial site. At this point, it is crucial to emphasize that, in the context of this study, permafrost plays a more significant role as a storage mechanism for peatland OM rather than serving as a unique source of OM by itself.

In Northwest and Central Europe, continuous permafrost prevailed at approximately 27-17 cal kyr BP (Vandenberghe and Pissart, 1993), with the deposits likely degrading and shrinking due to the warming observed at the end of the LGM. For instance, in the Netherlands, there has been evidence of widespread permafrost degradation between 22 and 21 cal kyr BP (Van Huissteden et al., 2000), with continuous permafrost in the Dutch coversand region completely disappearing after 20 cal kyr BP (Bateman and Van Huissteden, 1999). This episode is an example of permafrost thawing that may have contributed to wetland development between ca. 21 and 16 cal kyr BP as recorded in our $P_{aq}$ record (Figure 2d). Between 17 and 15 cal kyr BP, permafrost zones in the study region were restricted to areas near the retreating ice-sheets (Renssen and Vandenberghe, 2003, and references therein). Afterwards, apart from a short later period of discontinuous permafrost (ca. 10.9 - 10.5 cal kyr BP) that has also been reported for Northwest and Central Europe (Vandenberghe and Pissart, 1993), there is evidence of the presence of permafrost in this region during the Younger Dryas (e.g., Isarin, 1997; Petera-Zganiacz and Dzieduszyńska, 2017).

The erosion of European permafrost and peatland deposits by glacial meltwater is a mechanism likely to have exported OM to the ocean. The decay of the European ice sheets and glaciers at the onset of the last deglaciation (e.g., Marks, 2002; Rinterknecht et al., 2006; Ó Cofaigh and Evans, 2007; Ballantyne and Ó Cofaigh, 2017; Patton et al., 2017) contributed to strengthening the Channel River discharge into the Bay of Biscay (Antoine et al., 2003; Bourillet et al., 2003) and it has been proposed that this was the main mechanism controlling the river activity during this time period (Toucanne et al., 2009). Deglacial pulses of meltwater emanating from the BIIS (at 22 and 18.6 cal kyr BP) and the FIS (starting around 19 cal kyr BP) (Bowen et al., 2002; Rinterknecht et al., 2006) were routed to the North Atlantic via the Channel River. In this way, the activity of the river responded to changes in the European ice masses, being particularly influenced by the dynamics of the FIS (Toucanne et al., 2010).

Although subglacial meltwater can flow through permeable sediments as groundwater, the presence of frozen ground with reduced hydraulic transmissivity, i.e., permafrost, hinders this process (Piotrowski, 1997). This leads to trapped pressurized

water accumulating underneath ice sheets and eventually draining during catastrophic events. As the climate warmed and the ice sheets retreated and/or permafrost decayed, bursts of subglacial meltwater were released, carving glacial features known as tunnel valleys in the ground and discharging large amounts of eroded material into rivers (Piotrowski, 1994, 1999; Kirkham et al., 2022, and references therein). Subglacial channels from major Pleistocene glaciations are still present today in Europe and serve as evidence of this phenomenon (e.g., Piotrowski, 1999; Piotrowski et al., 1999). Meltwater streams from the FIS discharged through the Elbe River and provoked several flooding events of the Channel River (Mangerud et al., 2004; Toucanne et al., 2015), with remarkable episodes (R2-R5) recorded as peaks in the ratios Ti/Ca and Fe/Ca of both the GeoB23302-2 and the MD95 2002 archives (Supplementary Figure 3). Floods R2-R5 were most likely associated with enhanced processes of erosion and sediment export in the catchment (see e.g., Bogen and Bønsnes, 2003) and the intensified freshwater influx, resulting from riverine discharge due to a mixture of precipitation and glacial meltwaters, is reflected in the terrestrially-sourced OM signal shown by our BIT index record, which corroborates that reported for core MD95 2002 (Ménot et al., 2006) (Supplementary Figure 4). Furthermore, the process of post-glacial sea-level rise may have played a role in the erosion of coastal permafrost deposits, potentially serving as an additional pathway for the transport of OM to the ocean (e.g., Meyer et al., 2019).

The remobilization of OM from land to ocean is largely mediated by rivers, with factors such as precipitation and temperature being major regulators of fluvial C fluxes (Bauer et al., 2013, and references therein). Between 21 and 17 cal kyr BP, a temperature rise (Figure 2a) observed in various Atlantic environmental records (e.g., Arz et al., 1999; Bard et al., 2000; Combourieu Nebout et al., 2002; Pailler and Bard, 2002), marked a transition from cold and dry to warm and wet conditions in continental Europe. For example, a gradual increase in the Northeast Atlantic SST starting at about 25 cal kyr BP and preceding the peak of terrigenous deposition is recorded in core SU8118 (37°46'N, 10°11'W) (Bard et al., 2000). This same warming trend can be observed on land, reflected in the $\delta^{13}$C signature of a speleothem record from Western Europe (45°30'N, 0°50'E) (Wainer et al., 2011) and starting roughly at the same time (Figure 2b). The speleothem timeseries is not high resolution and long-term trends may be in fact punctuated with short-term oscillations. In any case, it is important to acknowledge that, although speleothem $\delta^{13}$C values can potentially serve as a proxy for temperature (see e.g., Lechleitner et al., 2021, and references therein), the correlation must be interpreted with caution due to several other factors influencing C isotopic ratios in these archives (Fohlmeister et al., 2020). Enhanced precipitation in response to warming led to increases in fluvial runoff and, due to widespread permafrost, increased erosion and transport of sediment from land to the Channel River outlet (Ménot et al., 2006). After approximately 18 cal kyr BP, as the climate warmed, the area occupied by peatlands in Europe increased (Müller and Joos, 2020). This is in agreement with our $P_{aq}$ index record, which indicates the re-establishment of formerly frozen wetlands, potentially including peat-rich environments (Figure 2d).

Considering that the OM buried in marine sediment is only a relatively small part of the total OM entering rivers, which is predominantly returned to the atmosphere as $CO_2$ (e.g., Aufdenkampe et al., 2011), the OM export to the Bay of Biscay via the Channel River is likely to have been accompanied by the transfer of $CO_2$ and $CH_4$ to the atmosphere (e.g., Schneider Von Deimling et al., 2015; Schuur et al., 2015; Bröder et al., 2018). It follows that our comprehensive analysis, encompassing biomarkers, elemental proxies, and radiocarbon dating, consistently corroborates the hypothesis of permafrost thawing in

the Northern Hemisphere contributing to the observed perturbations in the atmospheric C reservoir (Köhler et al., 2014). This essentially means that Northwest and Central Europe too, similar to other permafrost sites (Winterfeld et al., 2018; Meyer et al., 2019), may have contributed to the deglacial rise in atmospheric $CO_2$ (Köhler et al., 2014; Marcott et al., 2014). However, it is likely that the deglacial loss of European permafrost was offset by the subsequent accumulation of significant amounts of C in permafrost-free soils and peatlands (Lindgren et al., 2018). Our elevated pre-depositional ages at ca. 17.5 cal kyr BP (up to ca. 15 [14]C kyr) may partly explain the steep drop in the [14]C signature of atmospheric $CO_2$ during the period known as the Mystery Interval (17.5 – 14.5 kyr BP) (see e.g., Broecker and Barker, 2007). In other words, this result implies that thawing European permafrost combined with the deep ocean reservoir contributed [14]C-depleted $CO_2$ to the atmosphere during this period. However, our results show that the remobilization of C from this terrestrial pool started as early as ca. 20.2 cal kyr BP, which considerably precedes estimates of large permafrost contributions to atmospheric $CO_2$ between 17.5 and 15 kyr BP (Crichton et al., 2016). This suggests the need to investigate leads and lags in the permafrost carbon feedback.

After approximately 18 cal kyr BP, a re-routing of the Elbe-Weser system meant that FIS meltwater carrying ancient C was being delivered to the Norwegian Channel (Toucanne et al., 2010, and references therein). After 17 cal kyr BP, sea-level rise caused a shift of the shoreline, with the Bay of Biscay no longer being suitable to record terrestrial runoff during the Holocene (Lambeck, 1997). This is reflected in the sudden drop observed in the BIT index record (Figure 2f). Notably, although our data support what has been previously inferred for the study region (Ménot et al., 2006; Rostek and Bard, 2013; Soulet et al., 2013), the distinct timing for the discharge peak observed in this study compared to other sites may imply different mechanisms of C remobilization and these need to be further investigated. Indeed, factors such as the local hydrology and vegetation have been shown to play a role in the accumulation and degradation pathways of permafrost-influenced peatlands (Hugelius et al., 2020, and references therein).

Peat-forming wetlands remain an important source of terrestrial OM to the ocean and of $CH_4$ and $CO_2$ to the atmosphere, with flux rates likely increasing due to current warming (Freeman et al., 2001; Hodgkins et al., 2014). In high northern latitude wetlands, it has been shown that permafrost degradation leads to wetland shrinkage (Avis et al., 2011). In the tropics the situation is also critical, with anthropogenic (Moore et al., 2013) and natural (Schefuß et al., 2016; Garcin et al., 2022) factors contributing to the remobilization of pre-aged C from peatlands. Therefore, the release of large amounts of peat-derived OM described here for deglacial Europe has analogues in the present day and may be useful to inform future projections of permafrost peatland loss (e.g., Fewster et al., 2022), with our results advocating for the importance of better constraining the C cycle in wetlands.

## 5   Conclusions

To reconcile the great pre-depositional ages observed here with geochemical data that do not hint towards highly-degraded petrogenic material, we argue that the OM in core GeoB23302-2 is mostly derived from ancient continental peat deposits. During the last interglacial, peatlands were established in the European landscape. These deposits were widely distributed and were preserved in a frozen state throughout the last glaciation due to the widespread presence of permafrost. Over the course of

the last deglaciation, warming and episodes of ice-sheet retreat and associated flooding through the Channel River resulted in the erosion of these permafrost deposits, enhancing the downstream transport of sediment and mobilising ancient C to the core site. Our results indicate that during the period between 20.2 and 15.8 cal kyr BP, a substantial portion of the OM transported to the Bay of Biscay originated from ancient European peatlands. After approximately 17 cal kyr BP, our core location was not suitable for recording terrigenous inputs via the Channel River. Instead, the Norwegian Channel may have become the primary recipient of fluvially-discharged permafrost-derived C. It is possible that the emission of greenhouse gases resulting from the degradation of formerly frozen OM in European permafrost contributed to the rapid rise of approximately 30 ppm in the atmospheric $CO_2$ concentration between 17.5 and 16 cal kyr BP. However, further investigation is needed to accurately quantify the rates and magnitudes of the processes responsible for this contribution. This study provides empirical evidence of a cycle of peat formation during warm periods and long-term storage under colder conditions. Owing to the size of the C pool involved, such a mechanism is likely to increase atmospheric greenhouse gas concentrations, with important implications for Earth's climate. In this context, our results will be useful to better constrain the role of ancient C mobilization and the permafrost carbon feedback in climate models.

*Data availability.* Data generated in this study are freely available at https://doi.pangaea.de/10.1594/PANGAEA.954937

*Author contributions.* GM designed the study. GM, KZ and HG collected the sediment core. Funding for the project was acquired by GM and EQA. EQA and WW conducted the laboratory analyses with the support of JH, HG, TG and TT. EQA, GM, JH, HG and WW analyzed the data. EQA wrote the paper and generated the figures. All authors provided feedback on the paper.

*Competing interests.* The authors declare that they have no conflict of interest.

*Acknowledgements.* This research was funded by the Alexander von Humboldt Foundation via a post-doctoral fellowship granted to EQA. Thanks are also due to the Brazilian National Council for Scientific and Technological Development (CNPq) for the support provided to EQA. HG was funded by the German Science Foundation within the Cluster of Excellence EXC 2077 "The Oceans Floor – Earth's Uncharted Interface" (Project number 390741603). We thank Dr Enno Schefuß, Professor Kita Macario and Fernanda Mattos for helpful discussions that benefited this research. We are also grateful for the data provided by Dr Samuel Toucanne and Professor Fortunat Joos, and for the technical laboratory support offered by Elizabeth Bonk and Lea Phillips.

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
