# Peer review of "Deglacial export of pre-aged terrigenous carbon to the Bay of Biscay"

_Climate of the Past, 2023_

## Referee Comment (RC1)

**Review of Queiroz Alves et al., Deglacial export of pre-aged terrigenous carbon to the Bay of Biscay in Climate of the Past**

This manuscript describes a geochemistry record from a marine core in the Bay of Biscay that spans the last 24 ka. The focus of the paper is the changing organic matter sources to the marine record including large influxes of reworked terrestrial soils and possible petrogenic material. The authors use the paleoclimate and paleo-landscape literature from Northwest Europe to attribute changes in organic matter sources to permafrost thaw, glacial processes, and vegetation change in the former Channel River watershed. They authors assert that the nature, age, and timing of terrigenous organic matter fluxes support the idea that respired organic matter from thawing permafrost in Europe contributed to the 30 ppm rise in atmospheric $CO_2$ between 17.5 and 16 ka.

Full disclosure: I am by no means an expert in the geochemical analysis presented in the paper. I hope that a separate reviewer can comment on these methodological details and their interpretation. This being said, the indices the authors have chosen to use as tracers of organic matter sources seem appropriate. My main issues with the paper relate to some of the inferences the authors make about what their data mean for the landscape processes in Europe, and the atmospheric CO2 record. I also make several suggestions for improving the organization and writing of the paper.

Overall, I think these issues can be addressed with major revisions.

-Ben Gaglioti

March 30, 2023

**General Comments**

1) The Abstract is lacking specific information about the results presented in the paper and how these results are used to form the conclusions.

2) Generally, many of the take-home messages reported in the Conclusion section are not adequately presented earlier in the paper. Specifically, what about the geochemistry data indicates that the organic matter in the marine core was derived from permafrost soils? Along these lines, the authors state that, due to sea-level rise, the post-17 ka portion of the core is not suitable to record terrestrial inputs, but they continue to make inferences about the terrestrial environment using this core. I urge the authors to verify that all their conclusions are backed up by results, contextualized in the Discussion, and that each conclusion does not preclude the others.

3) The authors describe how their data supports the idea that respired organic matter from thawing permafrost contributed to the 30 ppm rise in atmospheric $CO_2$ between 17.5 and 16 ka. This is because they observe a rise in ancient terrigenous-derived material at this time, and that this likely only represents a

fraction of the carbon that was respired to the atmosphere while being laterally transferred from land. For this inference to remain in the manuscript, the authors need to include several pieces of relevant information in the paper:

    a. How much C is required to contribute to a 30 ppm rise in $CO_2$ and is the size of that flux consistent with the size of the C pool that was exposed to permafrost thaw in Europe? Or, do other permafrost C pools need to be brought in to explain this?

    b. The authors should explain why they think the terrestrial C in the marine core indicates high rates of respiration during lateral transfer if they found that this material was highly labile in the marine core. If the organic matter had been heavily degraded during lateral transport, then would it have been deposited onto the sea floor as a relatively recalcitrant organic matter fraction.

    c. Significant ice-core research has been dedicated to identifying the potential sources of the deglacial $CO_2$. This is highly relevant here because these records would indicate if ancient permafrost carbon contributed to the deglacial rise in $CO_2$ during the 17.5-16 ka period. Recent data suggest that the Southern Ocean was the main source of $CO_2$ (Bauska et al., 2016) and relatively young wetland C from tropical to northern hemisphere sources were the likely source of $CH_4$ rise (Dionisius et al., 2020). This literature should be cited in the manuscript, and the authors should explain how their marine core record and the inferences they make about the permafrost carbon feedback relates to their inferences.

4) The authors do not describe how relative sea level rise would significantly shift the depositional zone where the Channel River was depositing terrestrial material until the Conclusions Section of the paper. This needs to be discussed earlier. The authors should also reconsider why relative sea level change would have only affected these depositional processes around 17 ka and not before or after this time.

5) Several portions of the Discussion Section need to be reworked in a way that relates back to the results in the paper. They currently read as background information that is rarely linked back up with the marine core record. See detailed comments below.

6) I wonder if the authors can briefly describe how they interpret the geochemical indices in the Methods Section. As it reads now, the Methods describe what these indices are used for, but not how higher or lower values are interpreted. Knowing this would make it easier for the reader to understand how the Results section are eventually interpreted.

**Detailed Comments**

**Abstract:** *Here we investigate the mobilization of organic matter to the Bay of Biscay at the mouth of the Channel River, where an enhanced terrigenous input has been reported for the last glacial-interglacial transition.*

**Comment:** Do you mean previously reported? Or is it being reported in this manuscript?

Lines 6-7: *A suite of biomarker and isotopic analyses on a high-resolution sedimentary archive provided the first direct evidence for the fluvial supply of **immature** and ancient terrestrial organic matter to the core location.*

**Line 7 Comment:** Instead of 'immature', I think you mean 'labile'. Immature implies some kind of ontogenetic stage, and it is confusing because the reader does not understand if you are talking about the $^{14}C$ age or the degree of diagenesis of the organic matter. Change throughout the manuscript. Also change mention of 'mature' organic matter to 'recalcitrant'.

**Abstract:** *In the light of what has been reported for other regions with present or past permafrost conditions on land, this result points to the possibility of permafrost carbon export to the ocean, caused by processes that likely furthered the observed changes in atmospheric carbon dioxide.*

**Comment:** This is vague. Briefly describe what has been reported by these previous studies. For instance, do the LGM permafrost maps suggest that the watershed of this river had permafrost during MIS 2, but not during the deglacial?

Also, I think you mean something like: *'…on land, which suggests that postglacial warming enhanced the release of permafrost carbon into the ocean and may do so again elsewhere as warming accelerates in the future.'*

**Comment Line 12:** Instead of saying 'immense', use the actual estimates for how much C is stored in permafrost.

**Introduction:** *…covering the region from Poland through Germany, the Netherlands and Belgium into France and Great Britain, in areas where permafrost cover no longer extends.*

**Comment:** Recommend: *'covering much of central and western Europe, in areas where permafrost cover no longer exists.'*

Also, cite Figure 1 here.

**Line 28 Comment:** Is this Petrogenic material considered an alternative explanation for the elevated terrestrial biomarker data described above. If so, I recommend stating that

this is an alternative and potentially permafrost-independent flux of C during this same time. It sounds as though you are preparing the reader to introduce two competing hypotheses that you will test here with your core data. 1) Glacier-stream-derived petrogenic C sources, and 2) Permafrost-derived soil organic C. These hypotheses are never stated, but I think they could be. In any case, the authors should describe how this background information is relevant to the question being asked here and how the marine record might answer this question.

**Line 53-54:** *Together, our results led to the identification of ancient and immature OM, likely sourced from European permafrost.*

**Comment:** It seems out of place to describe the main conclusion here in the Introduction before any of the data that supports this conclusion is presented.

**Lines 81-83:** *Apart from our results, Figure 2 shows the NGRIP   18O record (Andersen et al., 2004) and a time series for atmospheric CO2 concentration (Köhler et al., 2017) (Figure 2a) as well as records for sea surface temperature (SST) in the North Atlantic Ocean (Bard et al., 2000) and   13C from European speleothems (Wainer et al., 2011) (Figure 2b).*

**Comment:** Listing the studies that are featured in the Figure should be in the Discussion and accompanied by some information on how they relate to the data presented in this study.

**Line 97:** *…petrogenic C, while OM in Holocene samples **are** mostly **have** marine origins.*

**Line 97 Comment:** Rewrite

**Line 112 & 115 Comment:** I think your description of 'Aquatic vegetation' is not appropriate here. Aquatic vegetation usually implies plants that are submerged or emergent under seasonal or perennial surface water. I think you mean 'wetland' or 'hydric' vegetation here. Also, wetland vegetation can often consist of vascular plants, so the sentence on Line 116 describing an increase in vascular plants replacing wetland vegetation does not make sense here.

**Line 114-118 Comment:** It seems that you are attributing the CPI results to both wetland and steppe-tundra vegetation types for the period from 21-17 ka. How is an wetland-dominated vegetation consistent with a steppe-tundra vegetation occurring at the same time.  Please explain whether you are talking about two different time periods, or how you can reconcile these two inferences.

**Line 118-120:** *Our CPIalk record also provides clues to the degree of preservation of the sedimentary OM and, therefore, degradation processes happening during transportation (Bröder et al., 2018).*

**Line 118-120 Comment:** This diagenesis is not exclusively occurring in transport. Even active layer soils that are underlain by permafrost can have significant respiration, which means that at least some of these degradation processes likely occurred prior to lateral transfer.

**Line 121-123:** *The signal of more mature OM fluvially transported to the continental shelf is detected in our CPIalk and f records, which reach relatively low values during the peak of deposition when compared to the Holocene (Figure 2e).*

**Line 121-123 Comment:** This sentence is unclear. I recommend stating the interpretation of the old OM. Older OM relative to what? Then in the next sentence, describe the interpretation of the low values.

**Line 126 Comment:** Briefly explain why this lack of correlation between the two indices mean that they can be used for terrestrial vegetation reconstructions. This is necessary for the non-expert to understand the inferences made here.

**Line 126 Comment:** Overall, this interpretation of the fBB and CPI to infer vegetation needs its own paragraph with both topic and concluding sentences that describe the salient points of this part of the Discussion.

**Lines 135-136:** *In other words, pre-aged compounds during the Holocene are likely to be the result of lateral transport in the ocean.*

**Lines 135-136 Comment:** Before making this conclusion, you need to rule out other possible mechanisms of old n-alkanes. What are the lines of evidence supporting this and not supporting other sources.

**Lines 136-138:** *The pre-depositional ages of some of the compounds present in core GeoB23302-2 are considerably greater than those previously attributed to permafrost-derived OM at other sites and at different timescales (e.g., Gustafsson et al., 2011; Winterfeld et al., 2018).*

**Lines 136-138 Comment:** Due to the large variability of organic matter residence time in both permafrost and non-permafrost soils, the age of reworked organic matter is not a good indicator of permafrost here.

**Lines 149-151:** *Indeed, similarly to what happens in the deep ocean, the C pool in deep permafrost deposits is isolated from the atmospheric input of newly formed C species, with its 14C content being only subjected to decay, leading to a reservoir effect.*

**Comment:** This is key to the interpretation. I think the authors want to introduce this idea in the Introduction Section before introducing it at this late stage of the paper.

**General Comment: Only at the end of section 4.1 did I fully realize what the common theme that these paragraphs were addressing. Recommendations for**

**Organizing Section 4.1:** Based on the last sentence of this section, I think you are discussing two main conclusions from the paper here. That there was a 'massive mobilization of terrestrial C', and that a lot of this reworked terrestrial C was peat-derived material. As it reads now, I am not sure how some of the data described in detail in this section relates back to these two key points. Therefore, I think you should simplify this section to provide the evidence for and potential caveats / evidence against these conclusions in two separate paragraphs. One that focusses on the relevant data and literature that allows you to say that the C was terrestrial. And the other that enables you to conclude that it was likely from terrestrial peats.

**Line 156:** *Wetlands are ecosystems that store C and release CO2 due to the decomposition of OM.*

**Comment:** Wetlands are also ecosystems that fix $CO_2$ *from* the atmosphere. As described here, they have a one-way flux of $CO_2$ release, which is not true. Also, the presence of wetlands underlain by permafrost in the mid-latitudes during MIS 2 does not alone suggest that these wetlands were significant contributors to the deglacial CO2 rise. This requires some *change* in the fluxes between the major Carbon pools, not just the presence of certain pools.

More generally, I think the permafrost inference is a little backwards here. You are using circumstantial evidence to suggest that the OM in the core was temporarily stored in permafrost. Instead, I think you want to be describing what data in the core support the idea that permafrost C is a main source of the core OM. The reader still has not learned what about the core data has allowed you make this inference.

Also, this paragraph starts off as discussing permafrost C sources during the deglacial, but then moves on to discuss the potential for sub-glacial peat, potentially from the Eemian period, to be another significant OM source in the core without finishing the discussion on permafrost. It is not clear why this transition occurs and what significance this discussion point has on the marine core results. I recommend breaking these discussion points up into separate paragraphs and being clear how they relate to the results you present here.

**Lines 156-179 Comment:** This long paragraph reads more like background information about the relevant study area without mention of how background information is relevant to the specific results presented here. Either mention this relevant information in the Introduction, or relate it to your results or interpretation here.

**Lines 180-196 Comment:** This paragraph also goes into detail on the paleoclimate record of NW Europe without providing the proper context of why these topics and records are being discussed and how they relate to the patterns observed in the core data presented here. After a large body of research is reviewed in this section the authors only say that these data is all : '…in agreement with the Paq index record…' The details of this agreement are not described. Specifically, what patterns in the

marine record and what interpretation of those patterns, agrees well with the body of literature reviewed here?

I also recommend synthesizing this literature in a way that distills it down to the relevant points of the interpretation of interest here. This will likely result in a more concise section on the paleoclimate and paleo-landscape history of this region.

---

## Referee Comment (RC3)

Anonymous Referee review of cp-2023-7

Deglacial export of pre-aged terrigenous carbon to the Bay of Biscay

This manuscript by Queiroz Alves et al. presents a marine record of organic biomarkers, stable isotopes, radionuclides, and elemental ratios from the Bay of Biscay to reconstruct the carbon cycling history of this site which is hypothesized to be primarily driven by post-glacial fluxes of relict, terrestrial organic matter from western and central Europe since 24 ka. The authors also use a Bayesian mixing model framework to quantitatively estimate the contributions of three organic carbon endmembers: marine biomass, terrestrial material formed < 50 kyr, and $^{14}$C-depleted (-1000‰) petrogenic material. Their results suggest that previously reconstructed flood events in the prehistoric Channel River were responsible for an increased flux of terrestrial organic matter (OM) into the Bay of Biscay, with the most significant episode occurring between 17.5 and 16.5 ka. The authors also use this evidence to suggest that some of the mobilized terrestrial organic matter was also released as $CO_2$, contributing to the rise in atmospheric concentrations observed during the period of major Channel River floods.

The methods used to produce the original data in this manuscript appear to be sound and the general structure of the text is well organized. However, there are a number of issues with the interpretations of the proxy records and modeling results generated that have significant implications for the main takeaways of this study. That being said, the findings within this manuscript have the potential to provide the scientific community with valuable paleoclimate insights as to how rapid permafrost thaw affects local and global carbon cycling dynamics and climate feedbacks. I provide comments about each issue below, which I think can be addressed with major revisions to the text.

**General Comments**

Proxy Interpretations: This study utilizes a number of biogeochemical proxies, including *n*-alkanes, *n*-alkanoic acids, GDGTs, hopanes, and elemental ratios to explore the carbon cycling history of this marine sediment core. However, it is often unclear to the reader how each proxy is being interpreted. In the methods section of the main text, the authors should include statements about how changes in each proxy value are interpreted in this study in addition to the references supporting them (i.e. "greater BIT index values are interpreted as an increased contribution of terrestrial organic matter (Hopmans et al., 2004)"). In Figure 2, it looks like most of the original data is already plotted such that positive changes in values are interpreted as an increase in the terrestrial organic matter signal. Perhaps the authors can annotate this in Figure 2 to help the reader understand the major trends plotted in this information-rich graphic.

CPI: As elaborated on in the specific comments below, the authors' interpretations of the Carbon Preference Index for sedimentary *n*-alkanes (CPI$_{Alk}$) simultaneously as a proxy for vegetation change and thermally/biologically degraded terrestrial material are confusing and not well supported by the referenced literature. I do not recommend interpreting CPI$_{Alk}$ with a range from 4 to 6 as a signal of changing vegetation in this record. All vegetation, both terrestrial and aquatic, that is modern/contemporaneous or unaffected by organic matter degradation has a CPI$_{Alk}$ value > 1 and the high variability of values within plant taxonomic groups and habitats do not make this proxy a reliable

indicator of vegetation source changes (Bush & McInerney, 2013). The authors' secondary interpretation, that $CPI_{Alk}$ being > 1 throughout the record suggests heavily degraded, petrogenic OM is not a significant component of this carbon cycling system, is much sounder. However, the overlapping plots of $CPI_{Alk}$ and fßß in Figure 2 can be misleading because $CPI_{Alk}$ shows minimal change in labile vs. recalcitrant carbon sources over time while fßß suggests a change in the amount of terrestrial organic matter export around 17 ka. To address this, the $CPI_{Alk}$ plot could be separated from fßß in Figure 2 or moved to the supplemental materials as a separate plot since the interpretations of the two records are substantially different.

Mixing Model Implementation: The authors should provide more details about how the MixSIAR model was used in this study and how the results support the key findings of this manuscript. The methods section only briefly mentions that a dual-isotope mixing model was used in this study without any mention of the endmembers involved until the end of the results section, with the rest of the information being in the supplemental text. The supplement is missing key descriptions of the MixSIAR settings used in the model runs, including prior structure and trophic discrimination factors, as stated in the specific comments below. Such settings can greatly impact the output of the model run (Stock et al., 2018) and their absence renders these mixing experiments non-replicable. In the main text, the mixing model results shown in Figure 3 are only referenced twice, once in the results section and once in the discussion, before the concluding statements. These model results should be more integrated into the discussion with how they compare to other proxy results generated in this study.

Petrogenic OM: The description of the petrogenic carbon endmember is not clear throughout the manuscript and appears to change between multiple sections. In the introduction, the authors spend an entire paragraph explaining how petrogenic OM sourced from carbon-rich sedimentary rocks may be an important source of [14]C-depleted OM that may mask sedimentary archives of changing permafrost export. Then in the discussion section 4.1, the authors use their results to explain how there is likely no rock-derived OM signal in the core, and that the [14]C-depleted endmember is actually lignite (brown coal); although the source of this lignite in western and central Europe is not explained. Shortly after, the authors explain that peat deposits, previously explained in this text to be the terrestrial OM endmember containing more [14]C than the petrogenic source, have also been preserved in western and central Europe since the last interglacial. In that case, why do the authors choose to interpret that the more mobile, [14]C-depleted endmember as lignite instead of peat that formed way before the LGM? In Figure 3, the modeled petrogenic OM/lignite contribution is as high as ~60% but it is unclear how lignite could be preferentially mobilized over peat or permafrost from the same region. The authors need to be more consistent throughout the text with defining endmembers as permafrost, or peat, or lignite because it becomes very unclear by the conclusions which endmembers are being interpreted.

**Abstract**

Line 5: Clarify that the location of the Bay of Biscay is off the coast of modern-day France in this abstract?

Line 6: I suggest rephrasing the start of the sentence to use more active voice, something like "we present a suite of biomarker and isotopic analyses…".

Line 6: I recommend listing the biomarkers used in this study or at least a couple of examples.

Line 8: Change "this result" to "our results".

**Introduction**

Lines 28-36: In this paragraph, the authors should clarify that there are notable bedrock formations in the western and central Europe that might function as a source of petrogenic OM.

Line 34-36: Can the authors include/reference an example study where distinguishing OM sources between petrogenic and permafrost was critical to the interpretation?

Line 37-51: I think that the section on the LGM history of the European landscape would make more sense, organizationally, as the 2$^{nd}$ paragraph in this introduction because similar concepts are discussed in the 1$^{st}$ paragraph. Perhaps switch the 2$^{nd}$ and 3$^{rd}$ paragraphs but keep lines 51-54 as the end of the introduction?

Line 51: Change "Here, organic biomarkers…" sentence to use active voice.

**Materials and Methods**

Lines 56-77: All equations for the various biomarker indices mentioned in this section should reference the supplemental text (i.e. $CPI_{alk}$; Eq. S1). Also, the authors should make a statement about each biomarker measurement being an original contribution of this study before describing the indices calculated using those biomarkers.

Line 58: The "e.g.," appears to be in the wrong location in this sentence. Is it supposed to begin the list of references in parentheses starting with "Dypvik and Harris, 2001"?

Line 64: Add ", respectively" at the end of the phrase "continental vegetation systems", since $P_{aq}$ is not used to reconstruct OM degradation in this study.

Line 65-67: Based on the equation for $P_{aq}$ listed in Eq. S2, wouldn't this ratio directly describe the predominance of mid-chain *n*-alkanes? I suggest adding a statement about the proxy is interpreted; that lower $P_{aq}$ values reflect a greater contribution of terrestrial vascular plants.

Line 68: The statement about $CPI_{alk}$ being an indicator of OM degradation was already made in line 64.

Line 69-70: The authors should clarify that the BIT index is calculated from GDGT abundances while f$\beta\beta$ is calculated from hopane abundances. There should also be statements about how higher/lower index values are interpreted for each one.

Line 71: Specify that MixSIAR is the Bayesian mixing model used in this study, according to the supplemental text, and reference Stock et al. (2018).

Lines 77: This statement about methodology details being in the supplement should be moved to the start of this methods sections/paragraph.

**Results**

Lines 79-99: The authors should include a statement about their *n*-alkanoic acid $^{14}$C age results in this section.

Line 79: It would be helpful to have a statement about the length of geologic time recorded in this sediment core, based on the age-depth model results.

Lines 81-83: The information about Figure 2 in this sentence is already in the Figure 2 caption where it is more appropriate.

Lines 93-94: The BIT index record shown in Figure 2f should be referenced in this sentence.

Line 95-97: The reference to Supplementary Figure 2 is confusing in this sentence because that figure does not show any results of the MixSIAR model runs, only how the tracer values of the endmembers compare to the sediment mixture, which were determined before the model was run. The authors should remove the reference to that supplemental figure and only reference Figure 3 as they have also done in the following sentence.

**Discussion**

Line 102: The authors should restate/re-summarize the findings of Ménot et al. (2006) for ease of comparison with the results of this study. Also clarify which original results directly support the findings of the referenced study.

Lines 112-114: Please explain how terrestrial wetlands are a source of aquatic plants producing shorter *n*-alkane chain-lengths as opposed to other vegetation sources that may be contributing longer-chain waxes later in the downcore record. Wetlands also contain vascular, terrestrial plants which are often attributed as the primary source of longer-chain waxes (Freimuth et al., 2019).

Lines 114-116: I disagree with this statement that a CPI value between 4 and 5, compared to ~6 later in the record (Figure 2), confirms an increased flux of aquatic plants. CPI is typically not recommended for reconstructing vegetation changes with the interpretation used in this study. In Bush and McInerney (2013) and He et al. (2020), both referenced in the methods section, the CPI of aquatic/submerged vegetation is greater, on average, than that of some terrestrial plant types, albeit with very high variability. In that case, the $P_{aq}$ and CPI records would be explaining opposite trends in vegetation source. Please clarify which references support CPI being interpreted as a proxy for vegetation change.

Lines 116-118: How does an arid steppe and tundra landscape correlate to a greater presence of wetlands with submerged aquatic vegetation? And is the implication that the development of woody biomes replaced wetlands with a more forested landscape in western and central Europe?

Lines 118-121: The wording of this sentence is confusing, please rewrite it.

Lines 118-123: Please clarify which period is being referred to as having more "mature OM fluvially transported". Also, how can lower CPI values be interpreted as being both from aquatic plants and

petrogenic sources during the same time period? I recommend using CPI to only infer the degree of organic matter degradation and not vegetation change since the former is much more robust.

Lines 121-127: This section starts by claiming that CPI are recording a signal of more mature OM but the proxy but then explain why CPI cannot be used for that purpose in this record. Also, Bush and McInerney (2013) only demonstrate that CPI between gymnosperms and angiosperms are statistically different, but that does not support the vegetation interpretation here. In general, plant CPI values within a given taxonomic growth form are too variable to interpret between groups.

Line 127-132: These sentences describing the difference between petrogenic and coal-derived OM should be a separate paragraph.

Line 133: This introduction to the compound-specific $^{14}$C results is difficult to understand. Perhaps the authors can include an additional statement saying that the interpretation of an "ancient origin" for terrigenous biomarkers is derived from their $^{14}$C ages being older than the modeled age vs. depth relationship for this core? See also comment on Figure 2f for clarifying the relationship between the core chronology and compound-specific ages.

Lines 134-136: Does the "recent" part of the record only refer to the Holocene as described in Line 136? Also, can clarifying point be made that at some point in this record, the Channel River ceases to transport terrestrial OM from the European mainland and, therefore, the $^{14}$C reservoir and transportation mechanisms must be different during and after the presence of the Channel River?

Line 139: Change to "…petrogenic contributions are commonly thought to be absent [of] *n*-alkanoic acids"? As in, petrogenic OM typically do not contain *n*-alkanoic acids.

Line 142: List the ranges of δ$^{13}$C values for the core and organic-rich rocks referenced to demonstrate how much the two datasets differ.

Lines 144-146: I am not sure how the mixing model results support the argument that there is not a significant contribution of a true petrogenic OM endmember when it is not part of the model framework to begin with. Also, it is unclear where peak OM deposition is shown in Figure 3. Each endmember contribution in Figure 3 is plotted as a percentage of the total OM so the actual flux change in mass or volume unit per time is not obvious here.

Lines 151-152: This paragraph leading up to the concluding statement here needs more references to the specific time periods when terrestrial OM increased, both from the Figure 3 mixing model results and the referenced literature.

Lines 156-160: These sentences seem to suggest that while wetlands store carbon in the landscape, they might be responsible for releasing more relict carbon from Europe upon their establishment at the end of the LGM. This seems contradictory and requires further explanation of the cited literature. The compound-specific $^{14}$C data in this paper only has one data point prior to the end of the LGM so it seems difficult to support these statements with the original findings presented here.

Lines 160-162: Is the term "peatlands" being used in this context, and throughout the manuscript in general, as a synonym for wetlands? If so, I recommend sticking with one term for the entire text and if not, the distinction between the two terms should be made clear early on.

Lines 165-167: If last interglacial peat deposits are widespread throughout the region that is exporting terrestrial, relict carbon via the Channel River, could they also be a source of $^{14}$C-depleted in the studied core? The authors should explore whether this is may or may not be the case.

Lines 180-196: The paragraph presents a lot of background on the evidence for the increased export of permafrost OM following the LGM but only the $P_{aq}$ record produced in this study is mentioned as corroborating with the other literature. How do the referenced paleoclimate records compare to the mixing model results from this paper?

Lines 200-204: What line of evidence is used (i.e. sedimentation rate, geochemical proxies) to support this statement about increased Channel River discharge at the core location in this study? Also, the references to "the core location", Antoine et al. (2003) and Bourillet et al. (2003), are somewhat confusing because the methods of this manuscript describe the core in question (GeoB 23303-2) to be original data. If the references are talking about a different core collected close by, then the authors should make that clear; perhaps even including it in Figure 1.

Lines 208-214: This information about subglacial meltwater should be in the introduction to provide the reader with more context early on about why the export of terrestrial OM to this core site may have changed over time.

Lines 214-217: This sentence about connecting peaks in the Ti/Ca and Fe/Ca ratios is very important to one of the key claims of this paper that core GeoB 23303-2 likely records Channel River flooding events which potentially export more pre-aged OM. In that case, I recommend that Supplementary Figure 3 be moved to the main text to readily illustrate this point.

Lines 225-227: Which results, specifically, support the hypothesis described?

Lines 229-231: How do changes in compound-specific $^{14}$C ages in this study correlate to changes in the total amount of exported relict OM when three endmembers are involved? As stated in a previous comment, the MixSIAR results presented in Figure 3 show the proportional contribution of each endmember, not the total amount of OM which would require the total OM content of this core to be analyzed and presented, too. Without this information, it could be argued that the amount of exported OM did not increase at 17.5 ka, only the $^{14}$C age of the *n*-alkanoic acids being mobilized. Plant-derived compounds, including *n*-alkanoic acids and *n*-alkanes, can be preserved in permafrost that formed prior to the LGM (Vonk et al., 2017) and even during multiple, previous interglacials (Jongejans et al., 2022). Therefore, this core site could be integrating a highly variable pool of compound-specific $^{14}$C, even if the amount exported is not significantly changing over time.

Lines 239-240: The authors previously attribute their $^{14}$C-depleted endmember to lignite, not degraded Eemian peatlands, which makes this statement confusing. Or was this supposed to say "Eurasian peatlands"?

**Conclusions**

Lines 260-261: Are European peatlands actually being interpreted as the $^{14}$C-deplated, petrogenic endmember throughout this study instead of lignite? In Figure 3, the OC_petro endmember exceeds 60%, not the OC_terr endmember, which is describing the $^{14}$C and $\delta^{13}$C signature of peatlands in the

supplemental text while OC_petro is based on $^{14}$C-deplated lignite. This is also the first quantitative mention of the mixing model results, which should be addressed much more in the discussion section before making a concluding statement using them.

**Figures**

Figure 1: In the labels for the yellow and red dots, I suggest adding text to note which one refers to this study. Or maybe adjust the symbology to make it clearer which core is being presented as original data.

Figure 2f: For the compound-specific $^{14}$C results plotted here, it is difficult to determine how their ages compare to the corresponding modeled age of the sediment from which they were extracted since the y-axis is in uncalibrated $^{14}$C kyrs while the x-axis ages are adjusted to the $^{14}$C Marine20 calibration curve. Perhaps these results could, instead, be presented as age offsets from the core chronology (i.e. Gaglioti et al., 2014).

Figure 2 Caption: Figure 2e is listed twice. The second mention should be corrected to "f:".

Figure 3: It would be helpful to have similar x-axis annotations for the geologic/climatic time periods as shown in Figure 2, especially since the x axes time scales are different between Figures 2 and 3. The bands showing major Channel River flooding events should be included in this figure, too.

**Supplemental Text**

Line S28-29: Reference the figures, both in the main text and supplement, where these data are reported.

Line S65: Clarify that both branched and isoprenoid GDGTs were analyzed and reported in this study to calculate BIT index values.

Line S82: Unclear what "ELEMENTAR" is referring to in the parentheses.

Line S91: How much core depth was integrated to have 100 g of sediment? Did the authors consider the depth/time being integrated for each sample when determining OCter$_{-bio}$ model input statistics?

Line S147: Do "temporal variations" refer to the standard deviation of $\Delta^{14}$C measurements in the model inputs? If so, please clarify that.

Line S159-161: This paragraph is missing a number of important details on how MixSIAR was implemented for this study. Other sedimentary applications of MixSIAR (i.e. Menges et al., 2020; Douglas et al., 2022) include information of whether trophic discrimination factors were applied, which prior structure was used, what Markov Chain Monte Carlo settings were used to reach model convergence, etc. As it stands, the MixSIAR runs for this study are not replicable based on the information provided in the text. I also recommend including a table in the supplement that displays the summary statistics (mean and standard deviation) for each endmember from at least one model run since these details for endmember $\Delta^{14}$C inputs are not specified elsewhere.

Figure S1: While the interpretation and units of the y-axis are described in the caption, the y-axis on the figure should be changed to something like "Depth (cm)" for ease of reading.

Figures S3-S5 Captions: It would be helpful to clarify in each caption that data from core GeoB23303-2 was produced in this study. Initially, it is unclear whether the other studies referenced in the captions refer to one or all of the core IDs mentioned.

**References cited in this review**

Antoine, P., Coutard, J. P., Gibbard, P., Hallegouet, B., Lautridou, J. P., & Ozouf, J. C. (2003). The Pleistocene rivers of the English Channel region. *Journal of Quaternary Science: Published for the Quaternary Research Association, 18*(3-4), 227-243.

Bourillet, J. F., Reynaud, J. Y., Baltzer, A., & Zaragosi, S. (2003). The 'Fleuve Manche': the submarine sedimentary features from the outer shelf to the deep-sea fans. *Journal of Quaternary Science: Published for the Quaternary Research Association, 18*(3-4), 261-282.

Bush, R. T., & McInerney, F. A. (2013). Leaf wax n-alkane distributions in and across modern plants: implications for paleoecology and chemotaxonomy. *Geochimica et Cosmochimica Acta, 117*, 161-179.

Douglas, P. M., Stratigopoulos, E., Park, S., & Keenan, B. (2022). Spatial differentiation of sediment organic matter isotopic composition and inferred sources in a temperate forest lake catchment. *Chemical Geology, 603*, 120887.

Freimuth, E. J., Diefendorf, A. F., Lowell, T. V., & Wiles, G. C. (2019). Sedimentary n-alkanes and n-alkanoic acids in a temperate bog are biased toward woody plants. *Organic Geochemistry, 128*, 94-107.

Gaglioti, B. V., Mann, D. H., Jones, B. M., Pohlman, J. W., Kunz, M. L., & Wooller, M. J. (2014). Radiocarbon age-offsets in an arctic lake reveal the long-term response of permafrost carbon to climate change. *Journal of Geophysical Research: Biogeosciences, 119*(8), 1630-1651.

He, D., Nemiah Ladd, S., Saunders, C. J., Mead, R. N., & Jaffé, R. (2020). Distribution of n-alkanes and their δ2H and δ13C values in typical plants along a terrestrial-coastal-oceanic gradient. *Geochimica et Cosmochimica Acta, 281*, 31-52.

Hopmans, E. C., Weijers, J. W., Schefuß, E., Herfort, L., Damsté, J. S. S., & Schouten, S. (2004). A novel proxy for terrestrial organic matter in sediments based on branched and isoprenoid tetraether lipids. *Earth and Planetary Science Letters, 224*(1-2), 107-116.

Jongejans, L. L., Mangelsdorf, K., Karger, C., Opel, T., Wetterich, S., Courtin, J., et al. (2022). Molecular biomarkers in Batagay megaslump permafrost deposits reveal clear differences in organic matter preservation between glacial and interglacial periods. *The Cryosphere, 16*(9), 3601-3617.

Menges, J., Hovius, N., Andermann, C., Lupker, M., Haghipour, N., Märki, L., & Sachse, D. (2020). Variations in organic carbon sourcing along a trans-Himalayan river determined by a Bayesian mixing approach. *Geochimica et Cosmochimica Acta, 286*, 159-176.

Ménot, G., Bard, E., Rostek, F., Weijers, J. W., Hopmans, E. C., Schouten, S., & Damsté, J. S. S. (2006). Early reactivation of European rivers during the last deglaciation. *Science, 313*(5793), 1623-1625.

Stock, B. C., Jackson, A. L., Ward, E. J., Parnell, A. C., Phillips, D. L., & Semmens, B. X. (2018). Analyzing mixing systems using a new generation of Bayesian tracer mixing models. *PeerJ, 6*, e5096.

Vonk, J. E., Tesi, T., Bröder, L., Holmstrand, H., Hugelius, G., Andersson, A., et al. (2017). Distinguishing between old and modern permafrost sources in the northeast Siberian land–shelf system with compound-specific δ 2 H analysis. *The Cryosphere, 11*(4), 1879-1895.

---

## Author Comment (AC1)

**Review of Queiroz Alves et al., Deglacial export of pre-aged terrigenous carbon to the Bay of Biscay in Climate of the Past**

This manuscript describes a geochemistry record from a marine core in the Bay of Biscay that spans the last 24 ka. The focus of the paper is the changing organic matter sources to the marine record including large influxes of reworked terrestrial soils and possible petrogenic material. The authors use the paleoclimate and paleo-landscape literature from Northwest Europe to attribute changes in organic matter sources to permafrost thaw, glacial processes, and vegetation change in the former Channel River watershed. They authors assert that the nature, age, and timing of terrigenous organic matter fluxes support the idea that respired organic matter from thawing permafrost in Europe contributed to the 30 ppm rise in atmospheric $CO_2$ between 17.5 and 16 ka.

Full disclosure: I am by no means an expert in the geochemical analysis presented in the paper. I hope that a separate reviewer can comment on these methodological details and their interpretation. This being said, the indices the authors have chosen to use as tracers of organic matter sources seem appropriate. My main issues with the paper relate to some of the inferences the authors make about what their data mean for the landscape processes in Europe, and the atmospheric CO2 record. I also make several suggestions for improving the organization and writing of the paper.

Overall, I think these issues can be addressed with major revisions.

-Ben Gaglioti
March 30, 2023

We sincerely appreciate the thorough review conducted by Ben. The valuable comments have certainly enhanced the content of our manuscript and have contributed to improvements in the presentation of our data and conclusions. Below we answer the comments.

**General Comments**

1) The Abstract is lacking specific information about the results presented in the paper and how these results are used to form the conclusions.

Because we have used several different proxies in the study, we opted to not mention them individually in the abstract in order to be concise. We have changed the text to provide a summary of the analytical approaches employed:

*We conducted a comprehensive suite of biomarker analyses (e.g., n-alkanes, hopanes, and n-alkanoic acids) and isotopic investigations (radiocarbon dating and $\delta^{13}C$ measurements) on a high- resolution sedimentary archive.*

Later in the abstract we present the conclusions derived from these findings:

*The present study provides the first direct evidence for the fluvial supply of immature and ancient terrestrial organic matter to the core location. Moreover, our results reveal the possibility of permafrost carbon export to the ocean, driven by processes such as deglacial warming and glacial erosion.*

The details of how the results are used to form conclusions can be found in the discussion section of the manuscript.

2) Generally,many of the take-home messages reported in the Conclusion section are not adequately presented earlier in the paper. Specifically, what about the geochemistry data indicates that the organic matter in the marine core was derived from permafrost soils? Along these lines, the authors state that, due to sea-level rise, the post-17 ka portion of the core is not suitable to record terrestrial inputs, but they continue to make inferences about the terrestrial environment using this core. I urge the authors to verify that all their conclusions are backed up by results, contextualized in the Discussion, and that each conclusion does not preclude the others.

As discussed in section 4.1, the geochemical proxies indicate a peat origin for the OM. Additionally, the radiocarbon dating results show that this is an ancient source. In section 4.2, we discuss how the preservation of peat deposits due to the presence of permafrost may have led to the aging of this OM. We also discuss how this pre-aged peat-derived OM reached the Bay of Biscay following deglacial permafrost degradation. We suggest permafrost soil as a source using circumstantial evidence, namely the fact that permafrost has been reconstructed to prevail in much of the river catchment during the LGM, while it is almost entirely (except for high mountain regions) absent there today. We added some explanation to this end at various locations in the text, e.g., in the introduction:

*Notably, the European deglaciation was marked not only by the decay of ice sheets but also by the permanent and complete loss of this permafrost cover (e.g., Vandenberghe and Pissart, 1993; Levavasseur et al., 2011; 45 Vandenberghe et al., 2012; Žák et al., 2012; Schaefer et al., 2014; Vandenberghe et al., 2014), which raises the possibility of permafrost-derived OM being deposited on the continental shelf at the mouth of the Channel River.*

We acknowledge that our original sentence in the conclusion section may have been unclear and caused confusion. Although sea-level rise made our core location unsuitable for recording terrigenous inputs via the Channel River, it is important to note that terrigenous OM, although in reduced quantities (see BIT index record), continued to be deposited at the core location. We have revised this part of the conclusion to make this clearer:

*After approximately 17 kcal BP, our core location was not suitable for recording terrigenous inputs via the Channel River. Instead, the Norwegian Channel may have become the primary recipient of fluvially-discharged permafrost-derived C.*

3) The authors describe how their data supports the idea that respired organic matter from thawing permafrost contributed to the 30 ppm rise in atmospheric $CO_2$ between 17.5 and 16 ka. This is because they observe a rise in ancient terrigenous-derived material at this time, and that this likely only represents a fraction of the carbon that was respired to the atmosphere while being laterally transferred from land. For this inference to remain in the manuscript, the authors need to include several pieces of relevant information in the paper:

a. How much C is required to contribute to a 30ppm rise in $CO_2$ and is the size of that flux consistent with the size of the C pool that was exposed to permafrost thaw in Europe? Or, do other permafrost C pools need to be brought in to explain this?

The release of permafrost carbon as a potential contribution to the deglacial $CO_2$ rise is widely discussed in the literature. Both modeling (Köhler et al., 2014, Crichton et al., 2016) and empirical studies (e.g., Tesi et al., 2016, Winterfeld et al., 2018, Wu et al., 2022) have been conducted, and we refer to these studies to place our discussion into context. For example, the question of how much C is required to explain the 30 ppm rise is answered in the modeling papers as well as in the study by Winterfeld et al.

Throughout our manuscript the hypothesis raised is that European permafrost could have played a role in the observed $CO_2$ rise, recognizing that it would not be the sole contributing factor. This is clear, for example, in the following excerpt from our text:

*This essentially means that Northwest and Central Europe too, similar to other permafrost sites (Winterfeld et al., 2018; Meyer et al., 2019), may have contributed to the deglacial rise in atmospheric $CO_2$ (Köhler et al., 2014; Marcott et al., 2014).*

b. The authors should explain why they think the terrestrial C in the marine core indicates high rates of respiration during lateral transfer if they found that this material was highly labile in the marine core. If the organic matter had been heavily degraded during lateral transport, then would it have been deposited onto the sea floor as a relatively recalcitrant organic matter fraction.

The reviewer points to an important aspect in all of the research directed towards understanding the permafrost carbon feedback, i.e., to determine the degree to which thawed OM released from degrading permafrost deposits is remineralized during transport and following deposition in receiving reservoirs (e.g., ocean sediments). Estimates of this remineralized fraction range from 2 to 66 %. Therefore, this important question cannot be easily solved, and we take our data only as indicators that the process of mobilization occurred. We also use published evidence that some degree of degradation happens after thaw, however without suggesting exact fractions of loss. Nonetheless, our data can provide some indications towards the processes.

During the peak of terrigenous deposition, the CPI and the fbb proxies show that the OM in the core has undergone some level of degradation, either prior to erosion, or during transport. However, it is challenging to make direct inferences about the fraction of OM that may have been respired or degraded during transportation. If significant respiration or degradation occurred during transport, it is possible that a portion of the OM did not reach the ocean floor and therefore would not be reflected in the deposited sediments. Relying solely on the OM in the core may not provide a complete representation of the OM released from European permafrost during the deglaciation. However, the core records the remobilization of terrigenous OM to the Bay of Biscay and it is known from previous studies (e.g., Schneider Von Deimling et al., 2015; Schuur et al., 2015; Bröder et al., 2018) that this fluvial transport of OM may be accompanied by $CO_2$ and $CH_4$ emissions to the atmosphere. This is stated in section 4.2:

*Considering that the OM buried in marine sediment is only a relatively small part of the total OM entering rivers, which is predominantly returned to the atmosphere as $CO_2$ (e.g., Aufdenkampe et al., 2011), the OM export to the Bay of Biscay via the Channel River is likely to have been accompanied by the transfer of $CO_2$ and $CH_4$ to the atmosphere (e.g., Schneider Von Deimling et al., 2015; Schuur et al., 2015; Bröder et al., 2018).*

c.  Significant ice-core research has been dedicated to identifying the potential sources of the deglacial $CO_2$. This is highly relevant here because these records would indicate if ancient permafrost carbon contributed to the deglacial rise in $CO_2$ during the 17.5-16 ka period. Recent data suggest that the Southern Ocean was the main source of $CO_2$ (Bauska et al., 2016) and relatively young wetland C from tropical to northern hemisphere sources were the likely source of $CH_4$ rise (Dionisius et al., 2020). This literature should be cited in the manuscript, and the authors should explain how their marine core record and the inferences they make about the permafrost carbon feedback relates to their inferences.

We have included this information and the suggested reference in the introduction.

4)  The authors do not describe how relative sea level rise would significantly shift the depositional zone where the Channel River was depositing terrestrial material until the Conclusions Section of the paper. This needs to be discussed earlier. The authors should also reconsider why relative sea level change would have only affected these depositional processes around 17 ka and not before or after this time.

We also mention the impact of sea-level rise in our core location towards the end of the discussion:

*After 17 kcal BP, sea-level rise caused a shift of the shoreline, with the Bay of Biscay no longer being suitable to record terrestrial runoff during the Holocene (Lambeck, 1997). This is reflected in the sudden drop observed in the BIT index record (Figure 2f).*

Our focus is to describe and investigate processes happening during the last deglaciation, when the mouth of the Channel River was located close to the core

location. After 17 kyr BP, this configuration no longer applies and, therefore, our record is not suitable to make any inferences about depositional processes in the Bay of Biscay. We do mention, however, that the Elbe-Weser system was re-routed, indicating that the depositional processes that are relevant for our study were no longer taking place in the English Channel.

5) Several portions of the Discussion Section need to be reworked in a way that relates back to the results in the paper. They currently read as background information that is rarely linked back up with the marine core record. See detailed comments below.

We have replied to the detailed comments below.

6) I wonder if the authors can briefly describe how they interpret the geochemical indices in the Methods Section. As it reads now, the Methods describe what these indices are used for, but not how higher or lower values are interpreted. Knowing this would make it easier for the reader to understand how the Results section are eventually interpreted.

We have included this in Figure 2.

**Detailed Comments**

**Abstract:** *Here we investigate the mobilization of organic matter to the Bay of Biscay at the mouth of the Channel River, where an enhanced terrigenous input has been reported for the last glacial-interglacial transition.*

**Comment:** Do you mean previously reported? Or is it being reported in this manuscript?

We mean that the phenomenon has been previously reported elsewhere. We have made the necessary revisions to clarify this in our abstract.

Lines 6-7: *A suite of biomarker and isotopic analyses on a high-resolution sedimentary archive provided the first direct evidence for the fluvial supply of **immature** and ancient terrestrial organic matter to the core location.*

**Line 7 Comment:** Instead of 'immature', I think you mean 'labile'. Immature implies some kind of ontogenetic stage, and it is confusing because the reader does not understand if you are talking about the $^{14}$C age or the degree of diagenesis of the organic matter. Change throughout the manuscript. Also change mention of 'mature' organic matter to 'recalcitrant'.

While mature/immature and recalcitrant/labile can be related terms, they describe different aspects of the diagenesis and stability of OM. In our manuscript, the concept of maturity, expressed by the words "mature" and "immature", refers to the degree to which OM has undergone chemical and physical changes over time during diagenetic and katagenetic processes. The concept of OM maturity is a central one in organic geochemistry and petroleum geochemistry. The words "labile" and "recalcitrant" would not convey the same meaning. For example, mature OM tends to be more resistant to

decomposition than immature OM, but it is not necessarily recalcitrant. To avoid confusion, we have made changes to the following sentence:

*Additionally, the presence of petrogenic, i.e., thermally-mature, material (Farrington and Tripp, 1977; Jeng, 2006) is another factor to consider when interpreting this record.*

**Abstract:** *In the light of what has been reported for other regions with present or past permafrost conditions on land, this result points to the possibility of permafrost carbon export to the ocean, caused by processes that likely furthered the observed changes in atmospheric carbon dioxide.*

**Comment:** This is vague. Briefly describe what has been reported by these previous studies. For instance, do the LGM permafrost maps suggest that the watershed of this river had permafrost during MIS 2, but not during the deglacial?

Also, I think you mean something like: *'...on land, which suggests that postglacial warming enhanced the release of permafrost carbon into the ocean and may do so again elsewhere as warming accelerates in the future.'*

There is evidence for the presence of permafrost in the region during the LGM, as discussed and referenced in the introduction. Currently, permafrost is absent, suggesting that permafrost degradation happened during the transition between the LGM and the Holocene. We have revised the sentence in the abstract to make it clearer.

**Comment Line 12:** Instead of saying 'immense', use the actual estimates for how much C is stored in permafrost.

We have added a sentence with this information.

**Introduction:** *...covering the region from Poland through Germany, the Netherlands and Belgium into France and Great Britain, in areas where permafrost cover no longer extends.*

**Comment:** Recommend: '*covering much of central and western Europe, in areas where permafrost cover no longer exists.*'

Also, cite Figure 1 here.

We have made the text more concise and added a citation to Figure 1.

**Line 28 Comment:** Is this Petrogenic material considered an alternative explanation for the elevated terrestrial biomarker data described above. If so, I recommend stating that this is an alternative and potentially permafrost-independent flux of C during this same time. It sounds as though you are preparing the reader to introduce two competing hypotheses that you will test here with your core data. 1) Glacier-stream-derived petrogenic C sources, and 2) Permafrost-derived soil organic C. These hypotheses are never stated, but I think they could be. In any case, the authors should

describe how this background information is relevant to the question being asked here and how the marine record might answer this question.

Yes. In the previous paragraph, we focused on the permafrost hypothesis and, in this paragraph, we state that petrogenic material is another plausible source of OM to the oceans during the last deglaciation, explaining the mechanisms through which this could have occurred. We have addressed the reviewer's suggestion, adding a sentence to clearly state both hypotheses. Following this, the next paragraph is devoted to describe the environmental context of the study region.

**Line 53-54:** *Together, our results led to the identification of ancient and immature OM, likely sourced from European permafrost.*

**Comment:** It seems out of place to describe the main conclusion here in the Introduction before any of the data that supports this conclusion is presented.

We have removed this sentence from the introduction.

**Lines 81-83:** *Apart from our results, Figure 2 shows the NGRIP  18O record (Andersen et al., 2004) and a time series for atmospheric CO2 concentration (Köhler et al., 2017) (Figure 2a) as well as records for sea surface temperature (SST) in the North Atlantic Ocean (Bard et al., 2000) and  13C from European speleothems (Wainer et al., 2011) (Figure 2b).*

**Comment:** Listing the studies that are featured in the Figure should be in the Discussion and accompanied by some information on how they relate to the data presented in this study.

We have already included the mentioned data in the discussion section to support our interpretations of the results. We have deleted this sentence.

**Line 97:** *...petrogenic C, while OM in Holocene samples* **are** *mostly* **have** *marine origins.*

**Line 97 Comment:** Rewrite

Thank you for spotting this. We have changed the text accordingly.

**Line 112 & 115 Comment:** I think your description of 'Aquatic vegetation' is not appropriate here. Aquatic vegetation usually implies plants that are submerged or emergent under seasonal or perennial surface water. I think you mean 'wetland' or 'hydric' vegetation here. Also, wetland vegetation can often consist of vascular plants, so the sentence on Line 116 describing an increase in vascular plants replacing wetland vegetation does not make sense here.

In the sentence starting in line 112, we discuss the results of the $P_{aq}$ index. As stated earlier in the manuscript, this proxy indicates the relative contributions of terrestrial vascular plants, algae, and macrophytes. The latter are aquatic plants so the term "aquatic plants" in the sentence is correct. Later in the same sentence, we link the

presence of aquatic plants to wetlands because macrophytes are often associated with wetland ecosystems.

In the sentence starting in line 115, we do not mention wetlands but rather aquatic and vascular plants. Here the idea is to make the distinction between aquatic and terrestrial plants so we replaced "vascular plants" with "terrestrial vascular plants" to make this clearer.

**Line 114-118 Comment:** It seems that you are attributing the CPI results to both wetland and steppe-tundra vegetation types for the period from 21-17 ka. How is an wetland-dominated vegetation consistent with a steppe-tundra vegetation occurring at the same time. Please explain whether you are talking about two different time periods, or how you can reconcile these two inferences.

The suggested scenario for permafrost degradation begins with a steppe-tundra environment, characterized by cold temperatures and sparse vegetation. As permafrost thaws due to climate change, the resulting increase in temperature causes the expansion of wetland areas. We have decided not to use the CPI results as a vegetation proxy, and this sentence was removed from the text.

**Line 118-120:** *Our CPIalk record also provides clues to the degree of preservation of the sedimentary OM and, therefore, degradation processes happening during transportation (Bröder et al., 2018).*

**Line 118-120 Comment:** This diagenesis is not exclusively occurring in transport. Even active layer soils that are underlain by permafrost can have significant respiration, which means that at least some of these degradation processes likely occurred prior to lateral transfer.

Here we are considering a scenario in which the OM is frozen and, therefore, preserved from degradation until it is remobilized following erosion along river banks and coasts, for example. It is assumed that any degradation that occurs prior to transportation is negligible compared to degradation that occurs during the process of transportation. This is why we consider degradation during transport to be the most relevant in this scenario. However, it is true that the low CPI values are likely not solely a result of degradation processes occurring during transport but instead indicate a more degraded or mature source. We have changed the sentence to:

*Our CPI$_{alk}$ record reflects the degree of degradation the sedimentary OM has undergone in its previous terrestrial reservoir or during transportation (cf. Bröder et al., 2018).*

**Line 121-123:** *The signal of more mature OM fluvially transported to the continental shelf is detected in our CPIalk and f records, which reach relatively low values during the peak of deposition when compared to the Holocene (Figure 2e).*

**Line 121-123 Comment:** This sentence is unclear. I recommend stating the interpretation of the old OM. Older OM relative to what? Then in the next sentence, describe the interpretation of the low values.

We assume that, by old/older, the reviewer means mature/more mature. More mature relative to the Holocene, which is exactly the interpretation of the low values. We have changed the sentence to improve clarity.

**Line 126 Comment:** Briefly explain why this lack of correlation between the two indices mean that they can be used for terrestrial vegetation reconstructions. This is necessary for the non-expert to understand the inferences made here.

In the Materials and methods section, we explain both indices. While both are indicators of OM degradation, the CPI can also be used for vegetation reconstructions. In cases where the two proxies provide conflicting results, factors other than the level of OM degradation may be influencing the CPI results. In any case, we have decided not to use the CPI record as a proxy for vegetation.

**Line 126 Comment:** Overall, this interpretation of the fBB and CPI to infer vegetation needs its own paragraph with both topic and concluding sentences that describe the salient points of this part of the Discussion.

Only the CPI can be used to infer vegetation types (see previous answer). However, as similar comments have also been raised by the other reviewers, we have decided to refrain from using CPI as an additional proxy for vegetation.

**Lines 135-136:** *In other words, pre-aged compounds during the Holocene are likely to be the result of lateral transport in the ocean.*

**Lines 135-136 Comment:** Before making this conclusion, you need to rule out other possible mechanisms of old n-alkanes. What are the lines of evidence supporting this and not supporting other sources.

There are two different types of processes that can cause organic compounds to appear pre-aged by several thousands of years (see Kusch et al. 2021). The first type of process is related to residence in intermediate reservoirs and subsequent transportation of compounds from their source to their final location, while the second type involves the addition of compounds from other sources, such as petrogenic inputs. However, the geochemical data presented in our manuscript does not support the latter.

Regarding the former, during the Holocene, the mouth of the Channel River was not located at the core location. As a result, pre-aged compounds found in this location were likely resuspended from another location on the shelf and subsequently redeposited there.

**Lines 136-138:** *The pre-depositional ages of some of the compounds present in core GeoB23302-2 are considerably greater than those previously attributed to permafrost-derived OM at other sites and at different timescales (e.g., Gustafsson et al., 2011; Winterfeld et al., 2018).*

**Lines 136-138 Comment:** Due to the large variability of organic matter residence time in both permafrost and non-permafrost soils, the age of reworked organic matter is not a good indicator of permafrost here.

An interesting aspect of our research is that the pre-depositional ages measured in our study are considerably greater than those of permafrost-derived OM found at other locations. However, it is noteworthy that the ages reported in these other studies are comparable to one another (up to ca. 10,000 $^{14}$C yr; see e.g., Gustafsson et al., 2011; Winterfeld et al., 2018). While we believe it is important to mention this point in the manuscript, we have revised the sentence to acknowledge the variability mentioned by the reviewer.

**Lines 149-151:** *Indeed, similarly to what happens in the deep ocean, the C pool in deep permafrost deposits is isolated from the atmospheric input of newly formed C species, with its 14C content being only subjected to decay, leading to a reservoir effect.*

**Comment:** This is key to the interpretation. I think the authors want to introduce this idea in the Introduction Section before introducing it at this late stage of the paper.

We believe that the discussion section is a more appropriate place for the mentioned text rather than the introduction. This is because it addresses the pre-aged nature of the OM, which is a result derived from our analyses. Placing this text in the introduction, before presenting our results, may confuse the reader. We have removed the reference to the deep ocean as it could lead to confusion too.

**General Comment: Only at the end of section 4.1 did I fully realize what the common theme that these paragraphs were addressing. Recommendations for Organizing Section 4.1:** Based on the last sentence of this section, I think you are discussing two main conclusions from the paper here. That there was a 'massive mobilization of terrestrial C', and that a lot of this reworked terrestrial C was peat-derived material. As it reads now, I am not sure how some of the data described in detail in this section relates back to these two key points. Therefore, I think you should simplify this section to provide the evidence for and potential caveats / evidence against these conclusions in two separate paragraphs. One that focusses on the relevant data and literature that allows you to say that the C was terrestrial. And the other that enables you to conclude that it was likely from terrestrial peats.

Our study confirms the findings of Ménot et al. (2006) regarding the deglacial peak of terrigenous OM, as indicated in the first sentence of this section. However, we further expand upon their work by employing additional proxies that enable us to explore the origins of this terrigenous OM. The discussion of the biomarker results is presented in the first paragraph of this section, while the second paragraph focuses on the analysis of radiocarbon dating results.

To address the reviewer's suggestion and make the discussion clearer, we have reorganized this section into three paragraphs. In the first paragraph, we discuss the proxies used to confirm the terrigenous OM deposition. The second paragraph discusses the proxies used to investigate the sources of this OM coming from land. Finally, the last paragraph focuses on the radiocarbon dating results.

**Line 156:** *Wetlands are ecosystems that store C and release CO2 due to the decomposition of OM.*

**Comment:** Wetlands are also ecosystems that fix $CO_2$ *from* the atmosphere. As described here, they have a one-way flux of $CO_2$ release, which is not true. Also, the presence of wetlands underlain by permafrost in the mid-latitudes during MIS 2 does not alone suggest that these wetlands were significant contributors to the deglacial CO2 rise. This requires some *change* in the fluxes between the major Carbon pools, not just the presence of certain pools.

We acknowledge the reviewer's point about the multiple fluxes of $CO_2$ in wetlands, but our intention was not to imply a one-way flux but rather to highlight the $CO_2$ release aspect in the context of OM decomposition. We have rephrased the sentence to make this clearer:

*Wetlands are dynamic ecosystems that fix $CO_2$ from the atmosphere, store C and contribute to the C cycle through various processes, including the decomposition of OM that releases $CO_2$ (Mitra et al., 2003).*

It is true that the presence of wetlands alone does not necessarily suggest their contribution to the deglacial $CO_2$ rise. Our statement aimed to highlight the need for further investigation into potential factors, including thawing permafrost, which may contribute OM to the deposition site. In the preceding discussion (section 4.1), we identified the presence of a wetland source. Subsequently, highlighting the occurrence of wetlands in the region during this time period (section 4.2) provides additional support for our findings from the previous section.

More generally, I think the permafrost inference is a little backwards here. You are using circumstantial evidence to suggest that the OM in the core was temporarily stored in permafrost. Instead, I think you want to be describing what data in the core support the idea that permafrost C is a main source of the core OM. The reader still has not learned what about the core data has allowed you make this inference.

This discussion of how the data from the core support the idea of permafrost C as a source is in the previous section.

Also, this paragraph starts off as discussing permafrost C sources during the deglacial, but then moves on to discuss the potential for sub-glacial peat, potentially from the Eemian period, to be another significant OM source in the core without finishing the discussion on permafrost. It is not clear why this transition occurs and what significance this discussion point has on the marine core results. I recommend breaking these discussion points up into separate paragraphs and being clear how they relate to the results you present here.

We opted not to break this discussion into separate paragraphs because the discussion does not distinguish between permafrost carbon sources and Eemian peat. Instead, it suggests the potential preservation of Eemian peat due to the presence of permafrost during the LGM, which was subsequently released during the last deglaciation. We have added information between brackets to make this clearer:

*Although the environmental conditions of the last glaciation were unfavorable for the development of peatlands, factors such as the formation of permafrost in Europe resulted in the long-term preservation of OM from older periods (e.g., frozen peat OM) (Treat et al., 2014).*

**Lines 156-179 Comment:** This long paragraph reads more like background information about the relevant study area without mention of how background information is relevant to the specific results presented here. Either mention this relevant information in the Introduction, or relate it to your results or interpretation here.

We divided the discussion into two sections focusing on i) the possible sources of the OM (section 4.1) and ii) the mechanisms responsible for OM remobilization (section 4.2). While the former is based on our data, the latter discusses the actual events that took place in the study region during the last deglacial and that support our inferences from the previous section (largely from previous literature). Given the number of proxies analyzed and the complexity of the topic, this sequential order of presenting information makes the text less convoluted and improves clarity.

**Lines 180-196 Comment:** This paragraph also goes into detail on the paleoclimate record of NW Europe without providing the proper context of why these topics and records are being discussed and how they relate to the patterns observed in the core data presented here. After a large body of research is reviewed in this section the authors only say that these data is all : '...in agreement with the Paq index record...' The details of this agreement are not described. Specifically, what patterns in the marine record and what interpretation of those patterns, agrees well with the body of literature reviewed here? I also recommend synthesizing this literature in a way that distills it down to the relevant points of the interpretation of interest here. This will likely result in a more concise section on the paleoclimate and paleo-landscape history of this region.

The interpretation of the $P_{aq}$ index was discussed before in the text:

*Values for the Paq proxy point to a major contribution of OM from aquatic plants between approximately 20.2 and 17.2 kcal BP, suggesting the presence of OM sourced from peat and wetland vegetation (Figure 2d).*

Here we are correlating this with what is known in terms of landscape evolution in this region:

*After approximately 18 kcal BP, as the climate warmed, the area occupied by peatlands in Europe increased (Müller and Joos, 2020). This is in agreement with our Paq index record, which shows the re-establishment of previously frozen peatlands (Figure 2d).*

Section 4.2 was dedicated to the discussion of the existing knowledge from previous literature regarding the landscape development of our study region. We use this information to support the direct evidence derived from our sediment core, as discussed in section 4.1.

---

## Author Comment (AC2)

Queiroz Alves and colleagues present a record of organic biomarkers and carbon isotopes from a sediment core from the Bay of Biscay, which provides observational evidence of post-glacial release of terrestrial carbon. Using source-fingerprinting of different biomarker indices and a dual-carbon isotope-based carbon mixing model, the authors deduce carbon remobilization from ancient and frozen peat during deglacial climate warming. Because permafrost/peat thaw processes may have released a larger amount of organic carbon at the time, the authors suggest that in-situ remineralization of the liberated organic matter may have contributed to the onset of the rise of atmospheric $CO_2$ concentrations around 17.5 kyr ago as seen in Antarctic ice cores.

The paper is overall well-written and structured logically. The topic is timely and important as climate change scientists seek to better understand the functioning and potential climate impact permafrost thaw and carbon release may have. The methods used in this study are sound and I am convinced that the work was carried out in a careful and rigorous manner. However, I see some weaknesses in the writing of the paper and its data interpretation, which I believe can be addressed during major revisions.

We express our gratitude to the reviewer for their thorough review of our paper. We appreciate their valuable comments, which will certainly help to improve our study. Below, we provide answers to address the referee's comments.

**Overarching comments**

Methods: Some of the methods are insufficiently explained in the main manuscript, and I would like the authors to include more information about the functioning and limitations of the different proxies used in their work. For instance, a big part of their story rests on a $^{13}C/^{14}C$ mixing model, which involves quite large uncertainties and assumptions and thus needs to be explained in detail in the main manuscript. Also, it is not really clear from the methods which data were generated in the present work and which originate from earlier work.

We have provided more information about our methods based on the specific reviewer's requests below.

Age model: This is a central part of this study, and much of the results and interpretation rest on this. From my understanding, the age model of the GeoB23302-2 is not published elsewhere, so I would ask the authors to provide more detail about the $^{14}C$ measurements of the planktic foraminifera, including a data table of conventional $^{14}C$ ages and resulting calibrated ages. I also have concerns about the choice of the Marine20 calibration curve and the potential impact of an additional marine reservoir effect in planktonic foraminifera. Particularly during the last glacial, there may have been a large difference in terms of ventilation between the ocean surface and the interior. Hence, I wonder if the reservoir is not over-estimated when applied to planktonic $^{14}C$ ages. I would like the authors to expand on this topic in more detail and include a concise statement in the methods section of the paper.

The age-depth model has not been published elsewhere. We have included more information about the model in the supplementary material, including a data table and the OxCal code necessary to reproduce the model.

Concerns about the use of Marine20 seem to be more significant in polar regions, as mentioned by Heaton et al. (2020):

*In higher-latitude polar regions, MRAs and critically their possible fluctuations over time are expected to be larger due to significantly increased variability in ocean ventilation and air-sea gas exchange mostly arising from changes in sea ice extent and differences in wind strength (Butzin et al. 2005). This is particularly likely to be the case during glacial periods.*

Between ca. 15 and 32 kcal BP, the larger global-average MRA observed in Marine20 compared to Marine13 is due to methodological improvements in the former. The MRA estimates incorporated in Marine20 are more realistic than those of previous curves. As stated in Heaton et al. (2023), the radiocarbon community does not recommend using Marine13 over Marine20 in any situation. We have added a short sentence about this in the supplementary material, where the methods for the construction of the age-depth model are described.

Carbon sources: There is much talk about petrogenic carbon throughout the paper, although the authors do not seem to find petrogenic carbon (but ancient peat) to be an important source to the sediment C. If petrogenic carbon is not important, I wonder why a petrogenic ($^{14}$C-free) carbon source is even quantified here (your dual isotope model suggests petrogenic carbon contributions up to 60%), which contradicts your discussion and conclusion. I also wonder about the use of D$^{14}$C values of *n*-alkanoic acids as a terr-OC end member, as they may follow different transport and aging dynamics than the bulk material. Furthermore, *n*-alkanoic acids are more degradation resistant than other organic compounds and may thus represent a pre-aged terrestrial carbon pool. Have you considered to distinguish between a contemporary ($^{14}$C-young) terrestrial OM source (e.g., soil OM) and strongly pre-aged ($^{14}$C-depleted, but not free of $^{14}$C) terrestrial sources that represents LGM (or pre-LGM) peat OM?

We start the manuscript by stating the possible sources of C to our marine sediment record, including petrogenic C. To test this possible contribution of a $^{14}$C-depleted petrogenic source, we defined an OC_petro end-member. However, in our mixing model, we used the isotopic values of ancient peat material (i.e., lignite, a type of coal) to represent this end-member. Our choices of end-members are clearly stated in the manuscript, but we acknowledge that having a "OC_petro" end-member may be the cause of confusion here. Therefore, we have renamed this end-member as "OC_fossil".

Dual-carbon isotope mixing models present the constraint of having only three end-members, despite the likelihood of multiple sources contributing to the deposit. The model is just an approximation and most likely does not adequately represent reality. It is important to note that all end-members have limitations. In addition, it would not be appropriate to use a contemporary terrestrial OM source as an end-member in our mixing model because the sources of C are not the same as in the period of interest, meaning that the isotopic composition is probably different too.

The reviewer is right in pointing out that *n*-alkanoic acids might not be representative of the whole continuum of organic materials present in natural soils and permafrost deposits. Indeed, these biomarkers may be older than more labile compounds such as organic acids, proteins, pigments or even lignin phenols. On the other hand, the advantage of using biomarker ages is that these materials can be unambiguously assigned to the terrestrial realm, which is of great importance when examining marine records receiving considerable amounts of young marine materials. We would like to stress, however, that long-chain fatty acids have been used as indicators for terrestrial organic matter in a large number of studies (see reviews by Eglinton et al., 2021 and Kusch et al., 2021), making the data comparable. We have revised the supplementary material to mention that the use of the fatty acids is an initial approximation.

**Minor points**

Abstract

Line 9: What kind of processes? Carbon release, degradation and $CO_2$ emissions?

The sentence reads "…result points to the possibility of permafrost carbon export to the ocean, caused by processes that likely furthered the observed changes in atmospheric carbon dioxide." Therefore, in this sentence we are referring to the processes that led to terrestrial carbon export to the ocean, preceding factors such as carbon release and $CO_2$ emissions from the affected carbon pools. We have revised the sentence to explicitly mention the relevant processes, i.e., deglacial warming and glacial erosion.

Line 12-13: It would be helpful to clarify add how thawing permafrost affects the global carbon cycle.

Thawing permafrost can affect the global carbon cycle in several ways. However, for the sake of conciseness, we have revised the sentence to mention the most immediate outcome of thawing permafrost, which is the release of greenhouse gases.

Line 17: Rising atmospheric levels?

We have revised the sentence, replacing "elevated" with "rising".

Line 43-45: This sentence reads a bit awkward. What is the "strong response from this system" specifically?

Tributaries of the Channel River responded to changes in precipitation patterns and the melting of ice sheets with enhanced amounts of water flow. We have revised the sentence to make this clearer.

Line 50: The continental shelf and what about the deeper ocean?

Since our core records the input of terrestrial OM to the continental shelf, at the mouth of the Channel River, we did not mention export to the deeper ocean here as it is not something we can assess in this study.

Methods

Line 57-60: Please explain briefly how these elemental ratios work; i.e. why Zr/Rb is thought to mirror grain size and Fe/Ca is a proxy for terrigenous sediments.

We have added a brief explanation about these proxies.

Line 68: Similar here; how does the CPI indicate OM degradation?

We have added a brief explanation about the use of this proxy to assess OM degradation.

Line 69: Also the BIT and fββ indices need to be explained a bit more in the main manuscript.

We have added a short explanation for both proxies in the main manuscript.

Results

Line 80-81: Isn't the age model based on foraminiferal $^{14}$C ages? What is the OxCal time series agreement index?

Yes, the chronology of the core is based on an age-depth model constructed with $^{14}$C ages obtained from planktic foraminifera. The details of radiocarbon analyses and model construction are given in the supplementary material. We have added a sentence at the start of the methods section in the main manuscript to briefly mention how the chronology of the core was established. The OxCal overall agreement index of 99% is given in the results and it shows the good agreement between the $^{14}$C data and the model.

Line 81-83: Please add more information about these reconstructions. Clarify that the NGRIP and $CO_2$ reconstructed are based on ice cores, and explain what the sea surface temperature record based on.

We have added this information to the text.

Line 95: What are "relatively high contributions"? A dominating fraction?

During the period of enhanced deposition, the contribution of terrestrial and fossil carbon to the samples is relatively high compared to other periods in the core. These carbon sources dominate the samples during this period, as shown in Figure 3. We have changed the sentence to:

*The results of our Bayesian mixing model (Figure 3) corroborate the BIT index record, showing that terrestrial and fossil C dominantly contributed to samples during the period of enhanced terrigenous deposition, while OM in Holocene samples is mostly marine.*

Discussion

Overall: Most of the data interpretation rests on concentrations and relative contributions of terrestrial compounds and OC fractions. Have the authors considered to estimate fluxes of the terrestrial OC fractions and compounds (e.g., of alkanes or the terrigenous OM fraction in g per sqm per year), which may provide an additional perspective to the paleo-variability of terrestrial carbon release. If not, I encourage the authors to do so.

We now present mass accumulation rates of terrigenous biomarkers in the supplementary material.

Line 104-106: Please explain how the Zr/Rb ratio or coarse-grained sediments are consistent or indicative of terrestrial material.

The Zr/Rb ratio is a proxy for grain size and the deposition of coarse-grained sediments is enhanced during periods of flood and intense fluvial activity. We have added this explanation to the text.

Line 107: How are these two periods different? Why not say 20.5-16.5 ka BP?

The sentence was written in this way to account for a BIT decrease at approximately 19 kcal BP. We have revised the sentence now to explicitly mention this.

Line 111: Heinrich events are not introduced or explained.

A brief definition has been added.

Line 127: Isn't the CPI by definition >1 or 1? It's not surprising the CPI values remain above 1 throughout the core in such a near-coastal environment.

Bray and Evans (1961) have measured values slightly below 1. In our study the values remain way above this value and values > 3 are interpreted as indicators for fresh biospheric material. If the OM peak observed in our record was caused by the deposition of petrogenic OM, the CPI values would reflect that and we would probably observe values closer to 1.

Line 144-146: A few lines above you provide a discussion about different processes (transport vs. petrogenic C) that may lead to high pre-depositional ages. Which of those processes/end members were included in the mixing model, and will the resulting C fractions not just be dependent on the end members chosen? Did you account for possible C aging during transport, which would probably affect the age of peat material?

It is unlikely that transport alone would account for the values we observe in our study, e.g., pre-depositional ages of several thousand $^{14}$C yr. Phenomena related to OM remobilization in the ocean are mentioned as a possible explanation for the presence of pre-aged compounds at the core location during the Holocene but not as the most relevant cause of their aging.

Using data from a single location precludes estimates of transit time, as we do not know what the OM age was at the point of origin. Moreover, it is difficult to account for

OM transport in our analyses because the various ways in which C cycles during this transit time are not completely understood. The mixing model works as a first approximation to identify the most probable sources for the OM, but it can certainly be improved as more information on the several processes of OM transformation and degradation becomes available.

Line 169: Do the authors mean that peat from older periods (e.g. the last interglacial) was frozen during the glacial? Perhaps add something like "OM from older periods (e.g., frozen peat OM)" to clarify this.

We have accepted the suggestion and changed the sentence accordingly.

Line 195: Could post-glacial sea level rise and erosion of terrestrial deposits also contributed to thermo-erosion of frozen deposits?

The deglacial erosion of coastal deposits by sea-level rise is a relevant phenomenon that has been observed in other locations (e.g., Meyer et al. 2019). We acknowledge this possibility in the discussion:

*Furthermore, the process of post-glacial sea-level rise may have played a role in the erosion of coastal permafrost deposits, potentially serving as an additional pathway for the transport of OM to the ocean (e.g., Meyer et al., 2019).*

Line 225-229: It may have contributed to the post-glacial carbon cycle perturbations but it could be good to mention that the loss of terrestrial carbon reservoirs was likely compensated by expansion of soil carbon stocks and vegetation growth at the time or thereafter (e.g., see Lindgren et al., 2018, https://www.nature.com/articles/s41586-018-0371-0).

We agree that it is important to mention how carbon stocks developed after the deglacial perturbation analyzed in our study, and we have added the information to the manuscript.

Line 233-235: What kind of leads and lags do we expect during permafrost carbon feedback? I would think that permafrost thawing leads to $CO_2$ emissions rather instantaneously, while the lateral transport of permafrost organic matter to rivers and the ocean may cause a lag, such that the 'peak signal' is seen delayed in the offshore sediment archive. So this would actually point to the opposite; that the post-glacial release of old peat/permafrost organic matter in central Europe did not affect atmospheric $CO_2$. So again, how certain are the authors about the correctness of their age-depth model (see my main comment)?

In the early stages of permafrost thawing, vegetation regrowth may have partly offset carbon release (see e.g., Schuur et al. 2015). Moreover, the OM decomposition and resulting $CO_2$ emissions from permafrost can be influenced by various environmental parameters (see e.g., Grosse et al. 2011). In the case of our study area, it is possible that these rates were further enhanced as climate warmed. Particles travelling downstream spend the majority of their total transit time at rest in transient floodplain storage (see Bradley and Tucker, 2013). Clastic particles have been shown to spend 4-5 orders of magnitude more time in storage than in motion (e.g., Repasch et al.

2020). So the reviewer is correct in suggesting that particles mobilized after permafrost thaw might arrive in the recipient marine reservoir with a time delay relative to the event of thawing. However, rapid erosional processes like river braiding and sea-level rise induced coastal erosion will supply particles at a high rate, albeit their carbon age might be old. It is this concept that is underlying our research approach.

While we have taken great care to construct a robust age-depth model, any model is inherently an estimate. Following established best practices in the field, we used the most up-to-date marine calibration curve and have provided a detailed description of the model, along with the necessary data and code, to enable readers to assess and reproduce our results. The agreement of the age-depth model as well as the result of the outlier model indicate that the measured foraminifera radiocarbon ages are represented well.

Line 236: How do the authors explain the sudden drop of terrestrial C discharge (as shown by the BIT index, Paq and the age of *n*-alkanoic acids). Could the river re-routing be related to that? It is not very clear from reading this paragraph.

At around 18 kyr BP, the Elbe-Weser system underwent a re-routing, followed by a sea-level rise after 17 kyr BP. As a consequence, the GeoB23302-2 core location no longer received the strong terrigenous signal observed previously, which may account for the sudden drop in the indicators measured in our study. We have revised the sentence to clarify this point.

Line 254-255: Why is there so much weight on petrogenic C in the paper if it is not one of the main sources?

Given the great pre-depositional ages measured in our study, petrogenic C could be a dominant source of terrigenous C in our core. This has been shown to be the case in other regions (e.g., Meyer et al. 2019) and, therefore, the manuscript was written to explore and test this possibility, which was discarded based on the geochemical signature of the organic matter in the core. However, since the conclusions are meant to summarize the findings of the paper, we mention petrogenic C here to acknowledge its potential to explain some of the results we observe. To address the referee's comment, we have made changes in the introduction: i) the discussion of petrogenic sources has been reduced and ii) ancient peats are now also mentioned as a potential source of aged OM.

Line 261-263: This is a quite complicated way to say that terrestrial/permafrost carbon release likely continued also after 17 kyr, although this was not recorded in your core. I suggest to write this more clearly.

It is true that the sentence as originally written suggested that although the export of terrigenous C to the ocean was not recorded in our core, it continued. However, it also indicated its routing. We have split the original sentence into two to make it less complicated.

Line 263: In the light of the findings and the offset between the terrestrial carbon peak and the period of $CO_2$ rise I would not call this 'likely'. I think that 'it is possible' reflects this and the underlying certainty better.

The first terrestrial carbon peak precedes the period of $CO_2$ increase, while the second peak coincides with the start of rising atmospheric $CO_2$ levels. Previous studies have shown a causal relationship between rising atmospheric $CO_2$ levels and degrading permafrost (e.g., Winterfeld et al. 2018), which is why we initially used the word "likely". However, we acknowledge that many of the mechanisms of carbon cycling from source to sink need to be better understood in this region, and therefore we have revised the text to use the word "possible" instead.

Line 267: And what are these important consequences? Please be more precise with these implications.

We have revised the sentence to specify that increases in atmospheric greenhouse gas concentrations are the factors that would affect Earth's climate in this context.

.

---

## Author Comment (AC3)

Anonymous Referee review of cp-2023-7
Deglacial export of pre-aged terrigenous carbon to the Bay of Biscay

This manuscript by Queiroz Alves et al. presents a marine record of organic biomarkers, stable isotopes, radionuclides, and elemental ratios from the Bay of Biscay to reconstruct the carbon cycling history of this site which is hypothesized to be primarily driven by post-glacial fluxes of relict, terrestrial organic matter from western and central Europe since 24 ka. The authors also use a Bayesian mixing model framework to quantitatively estimate the contributions of three organic carbon endmembers: marine biomass, terrestrial material formed < 50 kyr, and $^{14}C$-depleted (-1000‰) petrogenic material. Their results suggest that previously reconstructed flood events in the prehistoric Channel River were responsible for an increased flux of terrestrial organic matter (OM) into the Bay of Biscay, with the most significant episode occurring between 17.5 and 16.5 ka. The authors also use this evidence to suggest that some of the mobilized terrestrial organic matter was also released as $CO_2$, contributing to the rise in atmospheric concentrations observed during the period of major Channel River floods.

The methods used to produce the original data in this manuscript appear to be sound and the general structure of the text is well organized. However, there are a number of issues with the interpretations of the proxy records and modeling results generated that have significant implications for the main takeaways of this study. That being said, the findings within this manuscript have the potential to provide the scientific community with valuable paleoclimate insights as to how rapid permafrost thaw affects local and global carbon cycling dynamics and climate feedbacks. I provide comments about each issue below, which I think can be addressed with major revisions to the text.

We are very grateful for this very thorough review of our manuscript. In order to improve our paper, we have addressed the reviewer's comments as detailed below.

**General Comments**

Proxy Interpretations: This study utilizes a number of biogeochemical proxies, including *n*-alkanes, *n*- alkanoic acids, GDGTs, hopanes, and elemental ratios to explore the carbon cycling history of this marine sediment core. However, it is often unclear to the reader how each proxy is being interpreted. In the methods section of the main text, the authors should include statements about how changes in each proxy value are interpreted in this study in addition to the references supporting them (i.e. "greater BIT index values are interpreted as an increased contribution of terrestrial organic matter (Hopmans et al., 2004)"). In Figure 2, it looks like most of the original data is already plotted such that positive changes in values are interpreted as an increase in the terrestrial organic matter signal. Perhaps the authors can annotate this in Figure 2 to help the reader understand the major trends plotted in this information-rich graphic.

We have provided informative details in Figure 2 to assist readers in interpreting the various proxies effectively.

CPI: As elaborated on in the specific comments below, the authors' interpretations of the Carbon Preference Index for sedimentary $n$-alkanes (CPI$_{Alk}$) simultaneously as a proxy for vegetation change and thermally/biologically degraded terrestrial material are confusing and not well supported by the referenced literature. I do not recommend interpreting CPI$_{Alk}$ with a range from 4 to 6 as a signal of changing vegetation in this record. All vegetation, both terrestrial and aquatic, that is modern/contemporaneous or unaffected by organic matter degradation has a CPI$_{Alk}$ value > 1 and the high variability of values within plant taxonomic groups and habitats do not make this proxy a reliable indicator of vegetation source changes (Bush & McInerney, 2013). The authors' secondary interpretation, that CPI$_{Alk}$ being > 1 throughout the record suggests heavily degraded, petrogenic OM is not a significant component of this carbon cycling system, is much sounder. However, the overlapping plots of CPI$_{Alk}$ and f$_{ßß}$ in Figure 2 can be misleading because CPI$_{Alk}$ shows minimal change in labile vs. recalcitrant carbon sources over time while f$_{ßß}$ suggests a change in the amount of terrestrial organic matter export around 17 ka. To address this, the CPI$_{Alk}$ plot could be separated from f$_{ßß}$ in Figure 2 or moved to the supplemental materials as a separate plot since the interpretations of the two records are substantially different.

We agree with the reviewer's point, which aligns with feedback from other reviewers. Therefore, we decided not to use the CPI index as a proxy for vegetation change in the manuscript. Sentences related to such an interpretation for this proxy were deleted. We chose to retain the figure as it stands, as both proxies indicating maturity can be logically presented within the same panel.

Mixing Model Implementation: The authors should provide more details about how the MixSIAR model was used in this study and how the results support the key findings of this manuscript. The methods section only briefly mentions that a dual-isotope mixing model was used in this study without any mention of the endmembers involved until the end of the results section, with the rest of the information being in the supplemental text. The supplement is missing key descriptions of the MixSIAR settings used in the model runs, including prior structure and trophic discrimination factors, as stated in the specific comments below. Such settings can greatly impact the output of the model run (Stock et al., 2018) and their absence renders these mixing experiments non-replicable. In the main text, the mixing model results shown in Figure 3 are only referenced twice, once in the results section and once in the discussion, before the concluding statements. These model results should be more integrated into the discussion with how they compare to other proxy results generated in this study.

To maintain conciseness and readability, we have chosen not to include the full description of the model in the main manuscript. However, to address the referee's comment, we have included the description and code to run the model in the supplementary material. Our manuscript includes several proxies, which are mentioned and discussed throughout in a logical sequence. The mixing model is an additional analysis that corroborates the information derived from the proxies and, as such, it is referenced in the appropriate parts of the text for comparison with the other analyses.

Petrogenic OM: The description of the petrogenic carbon endmember is not clear throughout the manuscript and appears to change between multiple sections. In the introduction, the authors spend an entire paragraph explaining how petrogenic OM

sourced from carbon-rich sedimentary rocks may be an important source of $^{14}$C-depleted OM that may mask sedimentary archives of changing permafrost export. Then in the discussion section 4.1, the authors use their results to explain how there is likely no rock-derived OM signal in the core, and that the $^{14}$C-depleted endmember is actually lignite (brown coal); although the source of this lignite in western and central Europe is not explained. Shortly after, the authors explain that peat deposits, previously explained in this text to be the terrestrial OM endmember containing more $^{14}$C than the petrogenic source, have also been preserved in western and central Europe since the last interglacial. In that case, why do the authors choose to interpret that the more mobile, $^{14}$C-depleted endmember as lignite instead of peat that formed way before the LGM? In Figure 3, the modeled petrogenic OM/lignite contribution is as high as ~60% but it is unclear how lignite could be preferentially mobilized over peat or permafrost from the same region. The authors need to be more consistent throughout the text with defining endmembers as permafrost, or peat, or lignite because it becomes very unclear by the conclusions which endmembers are being interpreted.

In the light of results from other studies, which are referenced in the manuscript, we acknowledge that petrogenic sources can contribute organic carbon to marine sediment. Therefore, to account for this possibility in our study region, we analyze proxies (i.e., CPI and fbb) and include a petrogenic endmember in our mixing model. However, after a thorough examination of our results, we provide extensive discussion that leads us to dismiss this option. We do acknowledge that the terminology used can cause confusion and we have renamed the OC_petro endmember as OC_fossil. For this endmember we chose lignite because it represents a fossil (i.e., $^{14}$C-free) material with stable isotope values similar to those of peat. Effectively, this could be (sufficiently) ancient peat too so we decided to remove mentions to lignite from the discussion and refer only to ancient peats.

**Abstract**

Line 5: Clarify that the location of the Bay of Biscay is off the coast of modern-day France in this abstract?

We have changed our sentence to include the location of the Bay of Biscay.

Line 6: I suggest rephrasing the start of the sentence to use more active voice, something like "we present a suite of biomarker and isotopic analyses...".

We have changed the sentence accordingly.

Line 6: I recommend listing the biomarkers used in this study or at least a couple of examples. Line 8: Change "this result" to "our results".

We have modified this sentence as requested by Reviewer 1.

**Introduction**

Lines 28-36: In this paragraph, the authors should clarify that there are notable bedrock formations in the western and central Europe that might function as a source of petrogenic OM.

We included this information at the end of the paragraph.

Line 34-36: Can the authors include/reference an example study where distinguishing OM sources between petrogenic and permafrost was critical to the interpretation?

This is the case for the already-mentioned papers by Meyer et al. (2019) and Wu et al. (2022).

Line 37-51: I think that the section on the LGM history of the European landscape would make more sense, organizationally, as the 2$^{nd}$ paragraph in this introduction because similar concepts are discussed in the 1$^{st}$ paragraph. Perhaps switch the 2$^{nd}$ and 3$^{rd}$ paragraphs but keep lines 51-54 as the end of the introduction?

We agree that changing the order of the paragraphs may enhance the logical flow of the text and we have rearranged them accordingly.

Line 51: Change "Here, organic biomarkers..." sentence to use active voice.

We have changed the sentence as suggested.

**Materials and Methods**

Lines 56-77: All equations for the various biomarker indices mentioned in this section should reference the supplemental text (i.e. CPI$_{alk}$; Eq. S1). Also, the authors should make a statement about each biomarker measurement being an original contribution of this study before describing the indices calculated using those biomarkers.

We have incorporated the requested statement at the beginning of the section. We have also added references to the equations in the supplementary material.

Line 58: The "e.g.," appears to be in the wrong location in this sentence. Is it supposed to begin the list of references in parentheses starting with "Dypvik and Harris, 2001"?

The term "e.g.," is used here to provide an example of one application of the Zr/Rb ratio, which can serve as a proxy for river runoff, among other potential uses. To improve clarity, we have changed the sentence to:

*"Therefore, here we report the ratio Zr/Rb as an elemental measure of grain size, which has been used as a proxy for river runoff (Dypvik and Harris, 2001; Kylander et al., 2011; Wang et al., 2011; Wu et al., 2020)."*

Line 64: Add ", respectively" at the end of the phrase "continental vegetation systems", since $P_{aq}$ is not used to reconstruct OM degradation in this study.

We have made the requested change.

Line 65-67: Based on the equation for $P_{aq}$ listed in Eq. S2, wouldn't this ratio directly describe the predominance of mid-chain *n*-alkanes? I suggest adding a statement about the proxy is interpreted; that lower $P_{aq}$ values reflect a greater contribution of terrestrial vascular plants.

Mid- and short-chain n-alkanes are typically more prevalent in aquatic plants, whereas long-chain n-alkanes tend to be more abundant in terrestrial plants. This is captured by the $P_{aq}$ ratio, as stated in the materials and methods section of the manuscript:

*"…while the $P_{aq}$ reflects the predominance of long-chain n-alkanes in terrestrial vascular plants as opposed to algae and macrophytes, which primarily synthesize short- to mid-chain n-alkanes (Bianchi and Canuel, 2011)."*

We have added information on how the proxy values are interpreted directly to Figure 2.

Line 68: The statement about $CPI_{alk}$ being an indicator of OM degradation was already made in line 64.

We have made revisions to the text. First, we mention the potential applications of these proxies and reference studies that have employed them for these purposes. Later in the section, we provide explicit explanations as to why the proxies can be used for these purposes.

Line 69-70: The authors should clarify that the BIT index is calculated from GDGT abundances while fßß is calculated from hopane abundances. There should also be statements about how higher/lower index values are interpreted for each one.

We have included this information in the text. For the interpretation of the proxies, we have added details in Figure 2.

Line 71: Specify that MixSIAR is the Bayesian mixing model used in this study, according to the supplemental text, and reference Stock et al. (2018).

We have changed the sentence to include this information.

Lines 77: This statement about methodology details being in the supplement should be moved to the start of this methods sections/paragraph.

We have moved the statement to the beginning of the section.

**Results**

Lines 79-99: The authors should include a statement about their *n*-alkanoic acid $^{14}$C age results in this section.

We have rephrased the sentence about these results as follows:

*"The $^{14}$C ages of the long chain n-alkanoic acids varied from approximately 10 to 39 $^{14}$C kyr. When converted to pre-depositional age estimates, it is possible to observe that at the peak of our BIT record, around 18 kcal BP, compounds pre-aged by up to ca. 25,000 $^{14}$C yr were delivered to the continental shelf (Figure 2f). Pre-depositional ages broadly follow the BIT record, with younger compounds observed from the end of the BIT peak (ca. 16 kcal BP) towards the Holocene."*

Line 79: It would be helpful to have a statement about the length of geologic time recorded in this sediment core, based on the age-depth model results.

The GeoB23302-2 sediment core spans from approximately 25 to 4 kcal BP. We included the time span of the core at the beginning of this section.

Lines 81-83: The information about Figure 2 in this sentence is already in the Figure 2 caption where it is more appropriate.

We have removed this sentence.

Lines 93-94: The BIT index record shown in Figure 2f should be referenced in this sentence.

We have incorporated a reference to Figure 2f within this sentence.

Line 95-97: The reference to Supplementary Figure 2 is confusing in this sentence because that figure does not show any results of the MixSIAR model runs, only how the tracer values of the endmembers compare to the sediment mixture, which were determined before the model was run. The authors should remove the reference to that supplemental figure and only reference Figure 3 as they have also done in the following sentence.

We have cited Supplementary Figure 2 here to illustrate the model end-members that we have used. However, we agree that since these are not results of the model, Figure 3 should be referenced instead. We have changed the reference accordingly.

**Discussion**

Line 102: The authors should restate/re-summarize the findings of Ménot et al. (2006) for ease of comparison with the results of this study. Also clarify which original results directly support the findings of the referenced study.

We have made the requested changes.

Lines 112-114: Please explain how terrestrial wetlands are a source of aquatic plants producing shorter *n*-alkane chain-lengths as opposed to other vegetation sources that may be contributing longer-chain waxes later in the downcore record. Wetlands also contain vascular, terrestrial plants which are often attributed as the primary source of longer-chain waxes (Freimuth et al., 2019).

The input of aquatic vegetation to the OM (as indicated by the $P_{aq}$ index) alongside the identification of terrestrial OM (as indicated by the BIT index) suggests a potential wetland source. While it is true that vascular terrestrial plants can also be found in wetlands, the combination of terrestrial OM with indications of aquatic vegetation provides further evidence pointing towards a wetland origin. Wetlands, as continental ecosystems, are commonly associated with the growth of aquatic plants, making this interpretation plausible.

These wetland environments have been known to exhibit elevated abundances of mid-chain (23/25) *n*-alkanes, which contribute to the $P_{aq}$ index. While our $P_{aq}$ values may be lower than those found for "pure" floating and submerged plants, it is important to consider the integrated nature of our record and the contribution of other vascular higher plants present in wetlands, as mentioned by the reviewer.

Lines 114-116: I disagree with this statement that a CPI value between 4 and 5, compared to ~6 later in the record (Figure 2), confirms an increased flux of aquatic plants. CPI is typically not recommended for reconstructing vegetation changes with the interpretation used in this study. In Bush and McInerney (2013) and He et al. (2020), both referenced in the methods section, the CPI of aquatic/submerged vegetation is greater, on average, than that of some terrestrial plant types, albeit with very high variability. In that case, the $P_{aq}$ and CPI records would be explaining opposite trends in vegetation source. Please clarify which references support CPI being interpreted as a proxy for vegetation change.

Following the referee's suggestion, we have decided not to discuss the CPI record as a proxy for vegetation change in our manuscript. This sentence has been deleted.

Lines 116-118: How does an arid steppe and tundra landscape correlate to a greater presence of wetlands with submerged aquatic vegetation? And is the implication that the development of woody biomes replaced wetlands with a more forested landscape in western and central Europe?

The suggested scenario for permafrost degradation begins with a steppe-tundra environment, characterized by cold temperatures and sparse vegetation. As permafrost thaws due to climate change, the resulting increase in temperature causes the expansion of wetland areas. We have decided not to use the CPI results as a vegetation proxy, and this sentence was removed from the text.

Lines 118-121: The wording of this sentence is confusing, please rewrite it.

We have revised the sentence to improve clarity.

Lines 118-123: Please clarify which period is being referred to as having more "mature OM fluvially transported". Also, how can lower CPI values be interpreted as being both from aquatic plants and petrogenic sources during the same time period? I recommend using CPI to only infer the degree of organic matter degradation and not vegetation change since the former is much more robust.

To clarify the period we are referring to, we have rephrased the sentence to:

*"During the peak of terrigenous deposition, the signal of more mature OM fluvially transported to the continental shelf is detected in our CPI$_{alk}$ and fββ records, which reach relatively low values when compared to the Holocene (Figure 2e)."*

Following the referee's suggestion, we have decided not to discuss the CPI record as a proxy for vegetation change in our manuscript.

Lines 121-127: This section starts by claiming that CPI are recording a signal of more mature OM but the proxy but then explain why CPI cannot be used for that purpose in this record. Also, Bush and McInerney (2013) only demonstrate that CPI between gymnosperms and angiosperms are statistically different, but that does not support the vegetation interpretation here. In general, plant CPI values within a given taxonomic growth form are too variable to interpret between groups.

We have removed the interpretation of CPI as a vegetation proxy.

Line 127-132: These sentences describing the difference between petrogenic and coal-derived OM should be a separate paragraph.

We have removed the sentence.

Line 133: This introduction to the compound-specific $^{14}$C results is difficult to understand. Perhaps the authors can include an additional statement saying that the interpretation of an "ancient origin" for terrigenous biomarkers is derived from their $^{14}$C ages being older than the modeled age vs. depth relationship for this core? See also comment on Figure 2f for clarifying the relationship between the core chronology and compound-specific ages.

We have changed the first sentence of this paragraph to incorporate this information. However, the detailed explanation of the calculation method and the interpretation of a pre-depositional age are provided in the supplementary material.

Lines 134-136: Does the "recent" part of the record only refer to the Holocene as described in Line 136? Also, can clarifying point be made that at some point in this record, the Channel River ceases to transport terrestrial OM from the European mainland and, therefore, the $^{14}$C reservoir and transportation mechanisms must be different during and after the presence of the Channel River?

The most recent part of the record refers to the Holocene. We explicitly discuss compounds from this period here because they were deposited after the peak in terrigenous OM deposition. By this stage in the manuscript, the reader is already

aware, based on e.g., the BIT index results, that it is likely to have been changes in OM sources and pathways following the deglaciation. In the next section, we focus on the landscape development and further explore these changes.

We have moved the discussion on the compounds deposited during the Holocene to a later part of the paragraph.

Line 139: Change to "...petrogenic contributions are commonly thought to be absent [of] *n*-alkanoic acids"? As in, petrogenic OM typically do not contain *n*-alkanoic acids.

We have revised the sentence to enhance clarity while keeping its original meaning.

Line 142: List the ranges of $\delta^{13}C$ values for the core and organic-rich rocks referenced to demonstrate how much the two datasets differ.

We have included the requested information in the sentence.

Lines 144-146: I am not sure how the mixing model results support the argument that there is not a significant contribution of a true petrogenic OM endmember when it is not part of the model framework to begin with. Also, it is unclear where peak OM deposition is shown in Figure 3. Each endmember contribution in Figure 3 is plotted as a percentage of the total OM so the actual flux change in mass or volume unit per time is not obvious here.

As requested above, we provided the $\delta^{13}C$ values of the bulk samples and those of possible petrogenic sources in the region. Based on these values, it is clear that the bulk samples' isotopic signatures cannot be explained when this "true petrogenic OM end-member" is considered. Our choice of end-members, on the other hand, provides a suitable mixing polygon to investigate the sources of the OM in the core (see Figure S2). Moreover, the CPI and the fββ record do not support the incorporation of a fully petrogenic source in the model.

Figures 2 and 3 are in the same timescale and, therefore, the peak of OM deposition in Figure 3 can be easily found by referring to Figure 2f.

Lines 151-152: This paragraph leading up to the concluding statement here needs more references to the specific time periods when terrestrial OM increased, both from the Figure 3 mixing model results and the referenced literature.

We have included a reference to the exact time period under consideration in this sentence.

Lines 156-160: These sentences seem to suggest that while wetlands store carbon in the landscape, they might be responsible for releasing more relict carbon from Europe upon their establishment at the end of the LGM. This seems contradictory and requires further explanation of the cited literature. The compound-specific $^{14}C$ data in this paper only has one data point prior to the end of the LGM so it seems difficult to support these statements with the original findings presented here.

The perceived contradiction seems to arise from the dynamic nature of wetland ecosystems, where various processes of C cycling occur. Although, under stable conditions, C can persist in the wetlands over long periods, various factors can trigger the release of this C.

Peat area starts to increase towards the end of the LGM, but we observe additional increases during the deglaciation period (Figure 2d), where we have several compound-specific $^{14}$C data points. We have changed the sentence to clarify this.

Lines 160-162: Is the term "peatlands" being used in this context, and throughout the manuscript in general, as a synonym for wetlands? If so, I recommend sticking with one term for the entire text and if not, the distinction between the two terms should be made clear early on.

In our manuscript the term "peatlands" is used to refer to a specific type of wetland where peat accumulates. It is difficult to stick to one term over the other because the meanings are different and, depending on the instance, one term is preferred over the other. For example, the $P_{aq}$ ratio is defined for wetlands. However, one of the hopanes analyzed in our study is indicative of the presence of peat. We have reviewed the use of both terms throughout the manuscript to make sure they are used appropriately.

Lines 165-167: If last interglacial peat deposits are widespread throughout the region that is exporting terrestrial, relict carbon via the Channel River, could they also be a source of $^{14}$C-depleted in the studied core? The authors should explore whether this is may or may not be the case.

Yes – Eemian peat deposits preserved by the presence of permafrost during the last glacial period degraded during the deglaciation. This is exactly what we argue in several parts of our manuscript. See some examples below:

*"After approximately 18 kcal BP, as the climate warmed, the area occupied by peatlands in Europe increased (Müller and Joos, 2020). This is in agreement with our Paq index record, which shows the re-establishment of previously frozen peatlands (Figure 2d). Processes such as thermal and physical erosion of these deposits (see e.g., Sidorchuk et al., 2009, 2011) led to pre-aged material reaching the final burial site."*

*"To reconcile the great pre-depositional ages observed here with geochemical data that do not hint towards highly-degraded petrogenic material, we argue that the OM in core GeoB23302-2 is mostly derived from ancient continental peat deposits. During the last interglacial, peatlands were established in the European landscape…"*

To make this even clearer we have added the following sentence to the discussion:

*"We propose that the Eemian peats represent the primary source of fossil biomarkers transported to the Bay of Biscay."*

Lines 180-196: The paragraph presents a lot of background on the evidence for the increased export of permafrost OM following the LGM but only the $P_{aq}$ record produced in this study is mentioned as corroborating with the other literature. How do

the referenced paleoclimate records compare to the mixing model results from this paper?

The mixing model employed incorporates lignite or ancient peat as the OC_fossil end-member. This means that the model outputs indicating relatively high percentages of OC_fossil (fossil peat) and OC_terr (ancient peat) in comparison to OC_mar during the last deglaciation are aligned with the interpretation of expanded peatlands reflected in the $P_{aq}$ index.

Lines 200-204: What line of evidence is used (i.e. sedimentation rate, geochemical proxies) to support this statement about increased Channel River discharge at the core location in this study? Also, the references to "the core location", Antoine et al. (2003) and Bourillet et al. (2003), are somewhat confusing because the methods of this manuscript describe the core in question (GeoB 23303-2) to be original data. If the references are talking about a different core collected close by, then the authors should make that clear; perhaps even including it in Figure 1.

Antoine et al. (2003) and Bourillet et al. (2003) worked on a different core and used sedimentation rates and the analyses of sedimentary facies to examine fluvial activity. We have replaced "core location" with "Bay of Biscay".

Lines 208-214: This information about subglacial meltwater should be in the introduction to provide the reader with more context early on about why the export of terrestrial OM to this core site may have changed over time.

We moved the initial part of this paragraph to the introduction in order to better introduce the concept of glacial erosion earlier in the manuscript. We opted to maintain the details about subglacial meltwater, which is a more specific mechanism, in the discussion. This allows us to establish a connection with the subsequent description of flood episodes.

Lines 214-217: This sentence about connecting peaks in the Ti/Ca and Fe/Ca ratios is very important to one of the key claims of this paper that core GeoB 23303-2 likely records Channel River flooding events which potentially export more pre-aged OM. In that case, I recommend that Supplementary Figure 3 be moved to the main text to readily illustrate this point.

Given that these results, while original, do not present novel findings (as they confirm previous results from a nearby core), we have chosen to include this figure in the supplementary material.

Lines 225-227: Which results, specifically, support the hypothesis described?

The combination of all of our results supports this. Biomarkers and elemental proxies (BIT, Fe/Ca and Zr/Rb) point to enhanced terrigenous deposition during the deglaciation. In the same time period: (i) the $P_{aq}$ index shows an enhanced contribution of aquatic plants; (ii) a decreasing fββ ratio reflects the input of a compound commonly found in peat/lignite; (iii) the pre-depositional ages point to an ancient origin for the OM; (iv) the output of the mixing model shows that OC_terr and OC_fossil are the most relevant OM sources. We have changed the sentence to mention the multiple

lines of evidence:

*"It follows that our comprehensive analysis, encompassing biomarkers, elemental proxies, radiocarbon dating, and a mixing model, consistently corroborates the hypothesis of permafrost thawing in the Northern Hemisphere contributing to the observed perturbations in the atmospheric C reservoir (Köhler et al., 2014)."*

Lines 229-231: How do changes in compound-specific $^{14}$C ages in this study correlate to changes in the total amount of exported relict OM when three endmembers are involved? As stated in a previous comment, the MixSIAR results presented in Figure 3 show the proportional contribution of each endmember, not the total amount of OM which would require the total OM content of this core to be analyzed and presented, too. Without this information, it could be argued that the amount of exported OM did not increase at 17.5 ka, only the $^{14}$C age of the *n*-alkanoic acids being mobilized.

The reviewer is correct that the model estimates the proportional contribution of the different endmembers, and in order to show that indeed more OM from a different source accumulated during a peak, OM accumulation rates would be needed. Unfortunately, OC contents of the core are not available at this stage. Instead, we now present accumulation rates of biomarkers (e.g., those of higher land-plant derived long-chain *n*-alkanoic acids), which display dramatic changes over the deglaciation period.

Plant-derived compounds, including *n*-alkanoic acids and *n*-alkanes, can be preserved in permafrost that formed prior to the LGM (Vonk et al., 2017) and even during multiple, previous interglacials (Jongejans et al., 2022). Therefore, this core site could be integrating a highly variable pool of compound-specific $^{14}$C, even if the amount exported is not significantly changing over time.

This is correct in principle, and it is indeed what we suggest, i.e., that the relative contributions from ancient peats change dramatically over time. In addition, considering the accumulation rates of *n*-alkanoic acids, which are highly variable too, we present evidence for a drastic change in the supply of terrigenous material, both in quality and quantity.

Lines 239-240: The authors previously attribute their $^{14}$C-depleted endmember to lignite, not degraded Eemian peatlands, which makes this statement confusing. Or was this supposed to say "Eurasian peatlands"?

For this endmember we chose lignite because it represents a fossil (i.e., $^{14}$C-free) material with stable isotope values similar to those of peat. Effectively, this could be (sufficiently) ancient peat too so we decided to remove mentions to lignite from the discussion and refer only to ancient peats. OC_petro was renamed OC_fossil.

**Conclusions**

Lines 260-261: Are European peatlands actually being interpreted as the $^{14}$C-deplated, petrogenic endmember throughout this study instead of lignite? In Figure 3, the OC_petro endmember exceeds 60%, not the OC_terr endmember, which is

describing the $^{14}$C and $\delta^{13}$C signature of peatlands in the supplemental text while OC_petro is based on $^{14}$C-deplated lignite. This is also the first quantitative mention of the mixing model results, which should be addressed much more in the discussion section before making a concluding statement using them.

Here we are referring to ancient European peatlands, which potentially contain lignite. We have included the contribution of ancient peat material (OC_ter) in the conclusions. The first mention of the quantitative results from the mixing model can be found in the results section of the manuscript. In the subsequent discussion, we discuss the meaning of the relatively higher proportions of OC_ter and OC_fossil end-members in comparison to OC_mar.

**Figures**

Figure 1: In the labels for the yellow and red dots, I suggest adding text to note which one refers to this study. Or maybe adjust the symbology to make it clearer which core is being presented as original data.

To avoid overcrowding the figure and ensuring clarity, we have included a note in the caption stating that GeoB23302-2 is the core used in our study. The caption already indicates that MD95 2002 is from a previous study, providing the necessary context for both cores.

Figure 2f: For the compound-specific $^{14}$C results plotted here, it is difficult to determine how their ages compare to the corresponding modeled age of the sediment from which they were extracted since the y-axis is in uncalibrated $^{14}$C kyrs while the x-axis ages are adjusted to the $^{14}$C Marine20 calibration curve. Perhaps these results could, instead, be presented as age offsets from the core chronology (i.e. Gaglioti et al., 2014).

It appears that there might be a misconception on the reviewer's side. What is shown on this graph are compound-specific ages at the time of deposition. This means that the ages displayed here are already corrected for decay that happened after deposition and are essentially offsets from the sediment age. We refer the reviewer to the details in the supplementary material.

Figure 2 Caption: Figure 2e is listed twice. The second mention should be corrected to "f:".

This has been corrected.

Figure 3: It would be helpful to have similar x-axis annotations for the geologic/climatic time periods as shown in Figure 2, especially since the x axes time scales are different between Figures 2 and 3. The bands showing major Channel River flooding events should be included in this figure, too.

Figure 3 has been updated according to the referee's suggestions.

**Supplemental Text**

Line S28-29: Reference the figures, both in the main text and supplement, where these data are reported.

We have added a reference to Figure 2c in the main manuscript.

Line S65: Clarify that both branched and isoprenoid GDGTs were analyzed and reported in this study to calculate BIT index values.

We have included this information in the sentence.

Line S82: Unclear what "ELEMENTAR" is referring to in the parentheses.

We revised the sentence to improve clarity.

Line S91: How much core depth was integrated to have 100 g of sediment? Did the authors consider the depth/time being integrated for each sample when determining $OC_{ter\text{-}bio}$ model input statistics?

Approximately 3 cm. The depth intervals are given in the data table available at PANGAEA. Yes – the $\Delta^{14}C$ values of $OC_{ter\_bio}$ were taken from measurements conducted on *n*-alkanoic acids from the same sediment layers as the bulk samples.

Line S147: Do "temporal variations" refer to the standard deviation of $\Delta^{14}C$ measurements in the model inputs? If so, please clarify that.

No, in this context, "temporal variations" refers to the actual changes in $^{14}C$ content that have occurred in the ocean over time. What we mean is that we cannot apply the same OCmar_bio value to all samples because they are from different time periods.

Line S159-161: This paragraph is missing a number of important details on how MixSIAR was implemented for this study. Other sedimentary applications of MixSIAR (i.e. Menges et al., 2020; Douglas et al., 2022) include information of whether trophic discrimination factors were applied, which prior structure was used, what Markov Chain Monte Carlo settings were used to reach model convergence, etc. As it stands, the MixSIAR runs for this study are not replicable based on the information provided in the text. I also recommend including a table in the supplement that displays the summary statistics (mean and standard deviation) for each endmember from at least one model run since these details for endmember $\Delta^{14}C$ inputs are not specified elsewhere.

To make all the mentioned model parameters clear, we now provide the code for the model we ran in the supplementary material. We also included the summary statistics example requested by the referee.

Figure S1: While the interpretation and units of the y-axis are described in the caption, the y-axis on the figure should be changed to something like "Depth (cm)" for ease of reading.

Figure S1 has been updated according to the referee's suggestions.

Figures S3-S5 Captions: It would be helpful to clarify in each caption that data from core GeoB23303-2 was produced in this study. Initially, it is unclear whether the other studies referenced in the captions refer to one or all of the core IDs mentioned.

We have included this information in the captions.

**References cited in this review**

Antoine, P., Coutard, J. P., Gibbard, P., Hallegouet, B., Lautridou, J. P., & Ozouf, J. C. (2003). The Pleistocene rivers of the English Channel region. *Journal of Quaternary Science: Published for the Quaternary Research Association, 18*(3-4), 227-243.

Bourillet, J. F., Reynaud, J. Y., Baltzer, A., & Zaragosi, S. (2003). The 'Fleuve Manche': the submarine sedimentary features from the outer shelf to the deep-sea fans. *Journal of Quaternary Science: Published for the Quaternary Research Association, 18*(3-4), 261-282.

Bush, R. T., & McInerney, F. A. (2013). Leaf wax n-alkane distributions in and across modern plants: implications for paleoecology and chemotaxonomy. *Geochimica et Cosmochimica Acta, 117*, 161-179.

Douglas, P. M., Stratigopoulos, E., Park, S., & Keenan, B. (2022). Spatial differentiation of sediment organic matter isotopic composition and inferred sources in a temperate forest lake catchment. *Chemical Geology, 603*, 120887.

Freimuth, E. J., Diefendorf, A. F., Lowell, T. V., & Wiles, G. C. (2019). Sedimentary n-alkanes and n- alkanoic acids in a temperate bog are biased toward woody plants. *Organic Geochemistry, 128*, 94-107.

Gaglioti, B. V., Mann, D. H., Jones, B. M., Pohlman, J. W., Kunz, M. L., & Wooller, M. J. (2014). Radiocarbon age-offsets in an arctic lake reveal the long-term response of permafrost carbon to climate change. *Journal of Geophysical Research: Biogeosciences, 119*(8), 1630-1651.

He, D., Nemiah Ladd, S., Saunders, C. J., Mead, R. N., & Jaffé, R. (2020). Distribution of n-alkanes and their δ2H and δ13C values in typical plants along a terrestrial-coastal-oceanic gradient. *Geochimica et Cosmochimica Acta, 281*, 31-52.

Hopmans, E. C., Weijers, J. W., Schefuß, E., Herfort, L., Damsté, J. S. S., & Schouten, S. (2004). A novel proxy for terrestrial organic matter in sediments based on branched and isoprenoid tetraether lipids. *Earth and Planetary Science Letters, 224*(1-2), 107-116.

Jongejans, L. L., Mangelsdorf, K., Karger, C., Opel, T., Wetterich, S., Courtin, J., et al. (2022). Molecular biomarkers in Batagay megaslump permafrost deposits reveal clear differences in organic matter preservation between glacial and interglacial periods. *The Cryosphere, 16*(9), 3601-3617.

Menges, J., Hovius, N., Andermann, C., Lupker, M., Haghipour, N., Märki, L., & Sachse, D. (2020). Variations in organic carbon sourcing along a trans-Himalayan river determined by a Bayesian mixing approach. *Geochimica et Cosmochimica Acta, 286*, 159-176.

Ménot, G., Bard, E., Rostek, F., Weijers, J. W., Hopmans, E. C., Schouten, S., & Damsté, J. S. S. (2006). Early reactivation of European rivers during the last deglaciation. *Science, 313*(5793), 1623-1625.

Stock, B. C., Jackson, A. L., Ward, E. J., Parnell, A. C., Phillips, D. L., & Semmens, B. X. (2018). Analyzing mixing systems using a new generation of Bayesian tracer mixing models. *PeerJ, 6*, e5096.

Vonk, J. E., Tesi, T., Bröder, L., Holmstrand, H., Hugelius, G., Andersson, A., et al. (2017). Distinguishing between old and modern permafrost sources in the northeast Siberian land–shelf system with compound-specific δ 2 H analysis. *The Cryosphere, 11*(4), 1879-1895.

---

## Author Response (AR1)

The comprehensive responses to the reviewer's comments were previously uploaded. The manuscript underwent only two significant changes thereafter. First, we decided to remove the mixing model as it did not substantially contribute to our interpretation; the relevant information was already evident in the results of other analyses. Second, we relocated the methods section from the supplementary material to the main manuscript.

---

## Referee Report (RR1)

Anonymous Referee follow-up review of cp-2023-7
Deglacial export of pre-aged terrigenous carbon to the Bay of Biscay

The authors have sufficiently responded to all of my comments on the first submitted version of their manuscript, and I thank them for considering my suggestions in their latest, revised version. The added descriptions of geochemical proxy interpretations, both in the text and figures, greatly improve the reader's ability to follow the strong arguments presented by the authors for enhanced European permafrost/peat mobilization during the last deglaciation. I recommend that this manuscript be accepted after addressing several, very minor changes listed below, at the discretion of the editor.

**Suggested Corrections:**

Line 43: The authors should specify that permafrost development during the end of the LGM occurred around a large portion of the Channel River, as shown in Figure 1, to clearly introduce the key concept in this study that the Channel River was responsible for exporting large amounts of permafrost OC.

Line 83: The "n" in $n$-alkanes should be in italics.

Line 88: Specify that crenarcheol is an isoprenoid GDGT, as opposed to a branched GDGT as introduced earlier in this sentence.

Line 255: Does this sentence also imply that the C31$\alpha\beta$R hopane is not abundant in LGM permafrost either? If so, this fact should be restated later in the discussion, such as Line 290, because it is a critical piece of evidence for distinguishing the OM sources listed in this study. Or if that is not the case, perhaps state around Line 290 that permafrost is more important as a storage mechanism of peatland OM than as a unique source of OM by itself in this study.

Line 260: The authors should include the pre-depositional ages of this study and the referenced publications to quantitatively compare them in this sentence.

Figures 2, S1, S3, S4, S5 (and elsewhere in the manuscript text): Change the x-axis label from "kcal BP" to "cal kyr BP". Kcal is usually the abbreviation for kilocalories, not thousands of calibrated years.

---

## Author Response (AR2)

Anonymous Referee follow-up review of cp-2023-7
Deglacial export of pre-aged terrigenous carbon to the Bay of Biscay

The authors have sufficiently responded to all of my comments on the first submitted version of their manuscript, and I thank them for considering my suggestions in their latest, revised version. The added descriptions of geochemical proxy interpretations, both in the text and figures, greatly improve the reader's ability to follow the strong arguments presented by the authors for enhanced European permafrost/peat mobilization during the last deglaciation. I recommend that this manuscript be accepted after addressing several, very minor changes listed below, at the discretion of the editor.

We are very grateful for the valuable input provided by the reviewer, which certainly contributed to the improvement of our manuscript. Below we address the reviewer's minor comments.

**Suggested Corrections:**

Line 43: The authors should specify that permafrost development during the end of the LGM occurred around a large portion of the Channel River, as shown in Figure 1, to clearly introduce the key concept in this study that the Channel River was responsible for exporting large amounts of permafrost OC.

This information has been incorporated into the introduction (L42-45).

Line 83: The "n" in $n$-alkanes should be in italics.

This has been fixed.

Line 88: Specify that crenarcheol is an isoprenoid GDGT, as opposed to a branched GDGT as introduced earlier in this sentence.

This is now specified in the text (L88).

Line 255: Does this sentence also imply that the C31$\alpha\beta$R hopane is not abundant in LGM permafrost either? If so, this fact should be restated later in the discussion, such as Line 290, because it is a critical piece of evidence for distinguishing the OM sources listed in this study. Or if that is not the case, perhaps state around Line 290 that permafrost is more important as a storage mechanism of peatland OM than as a unique source of OM by itself in this study.

In order to clarify this, we have added the following sentence to section 4.2:

*At this point, it is crucial to emphasize that, in the context of this study, permafrost plays a more significant role as a storage mechanism for peatland OM rather than serving as a unique source of OM by itself.*

Line 260: The authors should include the pre-depositional ages of this study and the referenced publications to quantitatively compare them in this sentence.

This information has been added to the text.

Figures 2, S1, S3, S4, S5 (and elsewhere in the manuscript text): Change the x-axis label from "kcal BP" to "cal kyr BP". Kcal is usually the abbreviation for kilocalories, not thousands of calibrated years.

This has been corrected throughout the text.